# Tuberculosis notifications in regional Victoria, Australia: Implications for public health care in a low incidence setting

Nompilo Moyo[1,2]*, Ee Laine Tay[3], James M. Trauer[1,4], Leona Burke[1], Justin Jackson[5,6], Robert J. Commons[7,8], Sarah C. Boyd[9], Kasha P. Singh[10], Justin T. Denholm[1,10]

1 Victorian Tuberculosis Program, Melbourne Health, Melbourne, Victoria, Australia, 2 School of Nursing and Midwifery, La Trobe University, Melbourne, Victoria, Australia, 3 Communicable Diseases Epidemiology and Surveillance Unit, Health Protection Branch, Public Health Division, Department of Health, Melbourne, Victoria, Australia, 4 School of Public Health and Preventive Medicine, Monash University, Melbourne, Victoria, Australia, 5 Department of Medicine, Albury-Wodonga Health, Wodonga, Victoria, Australia, 6 Faculty of Medicine, University of New South Wales Rural Clinical School, Albury Campus, Albury, New South Wales, Australia, 7 Internal Medicine Services, Ballarat Health Services, Ballarat, Victoria, Australia, 8 Global and Tropical Health Division, Menzies School of Health Research and Charles Darwin University, Darwin, Northern Territory, Australia, 9 Royal Brisbane and Women's Hospital, Brisbane, South Australia, Australia, 10 Department of Infectious Diseases, Doherty Institute for Infection and Immunity, University of Melbourne, Parkville, Victoria, Australia

* nompiloo@yahoo.com

**Data Availability Statement:** Data is available on this website: http://hdl.handle.net/11343/326349.

**Funding:** The authors received no specific funding for this work.

## Abstract

### Background

Regionality is often a significant factor in tuberculosis (TB) management and outcomes worldwide. A wide range of context-specific factors may influence these differences and change over time. We compared TB treatment in regional and metropolitan areas, considering demographic and temporal trends affecting TB diagnosis and outcomes.

### Methods

Retrospective analyses of data for patients notified with TB in Victoria, Australia, were conducted. The study outcomes were treatment delays and treatment outcomes. Multivariable Cox proportional hazard model analyses were performed to investigate the effect of regionality in the management of TB. Six hundred and eleven (7%) TB patients were notified in regional and 8,163 (93%) in metropolitan areas between 1995 and 2019. Of the 611 cases in the regional cohort, 401 (66%) were overseas-born. Fifty-one percent of the overseas-born patients in regional Victoria developed TB disease within five years of arrival in Australia. Four cases of multidrug-resistant tuberculosis were reported in regional areas, compared to 97 cases in metropolitan areas. A total of 3,238 patients notified from 2012 to 2019 were included in the survival analysis. The time follow-up for patient delay started at symptom onset date, and the event was the presentation to the healthcare centre. For healthcare system delay, follow-up time began at the presentation to the healthcare centre, and the event was commenced on TB treatment. Cases with extrapulmonary TB in regional areas

**Competing interests:** The authors have declared that no competing interests exist.

have a non-significantly longer healthcare system delay than patients in metropolitan (median 64 days versus 54 days, AHR = 0.8, 95% CI 0.6–1.0, P = 0.094).

## Conclusion

Tuberculosis in regional Victoria is common among the overseas-born population, and patients with extrapulmonary TB in regional areas experienced a non-significant minor delay in treatment commencement with no apparent detriment to treatment outcomes. Improving access to LTBI management in regional areas may reduce the burden of TB.

## Background

Early diagnosis and treatment of tuberculosis are crucial to the effectiveness of TB control programmes [1]. Previous studies have reported delayed TB diagnosis in countries with low incidence [2–4]. For example, Labuda et al. [4] conducted a study involving 21 patients and reported that eight (38%) had their TB diagnosis delayed by months. In a comparable study involving 34 patients with TB, Kelly et al. [3] reported that 17 (50%) were diagnosed with other diseases, and the average time between admission and tuberculosis diagnosis was five days. Some studies have explored the factors associated with delays in TB treatment [5–7]. In a systematic review and meta-analysis of 45 studies, male patients and extended travel times/distances to the initial healthcare provider were associated with shorter patient and provider delays [5]. In addition, unemployment, low income, haemoptysis, and positive sputum smears were associated with patient delay [5]. Another study involving 133 patients reported that cough and hospital admission were associated with shorter health system delays, while age 65 or older was associated with longer delays [7].

In a retrospective study involving 239,532 patients, Wallace et al. [8] found that TB prevalence and trends were comparable in metropolitan and regional areas. In a similar study including 16,784 patients, Abubakar et al. [9] compared the incidence of tuberculosis and treatment outcomes for patients living in metropolitan and regional areas from 2001 to 2003 and reported that 45% of cases did not complete therapy in regional areas, compared to 26% in metropolitan areas.

TB incidence in Victoria remains low, with 436 TB cases notified in 2018, representing 6.9 cases per 100,000 population [10]. In Australia, research to date has tended to focus on metropolitan areas, where case numbers typically predominate. Understanding TB treatment delays among regional patients provides important insights into VTP performance and is a critical step towards tuberculosis elimination. Globally, TB surveillance data have been recognised as an important data source for assessing the disease burden and epidemiological trends in TB [11]. Evaluating treatment outcomes and delays in regional areas will inform practice and policy. We aimed to describe notified TB cases in regional Victoria from 1995 to 2019, including trends and outcomes over these 24 years.

## Methods

### Data source

We used routinely collected TB surveillance data. Data for all notified active tuberculosis cases in Victoria are collected by the VTP nurse consultants. Data are stored electronically in the Public Health Events Surveillance System (PHESS). PHESS is a centralised surveillance

database containing data on all notifiable diseases in Victoria since 1991 [10]. The notification of active tuberculosis cases is mandatory in Victoria under the Public Health and Wellbeing legislation [10]. PHESS has standardised data collection templates to ensure consistency. Nurse consultants record patient demographic, clinical data and TB contacts in PHESS.

Data on the estimated resident population for all local government areas were obtained from the Australian Bureau of Statistics (ABS). The estimated resident population is the official figure of Australia's population based on the concept of "usual residence" and refers to all people, regardless of nationality or citizenship, who usually live in Australia, except foreign diplomatic personnel and their families [12].

**Study design.** We conducted a respective cohort study. Patients notified to the Australian department of health with active TB from 1995 to 2019 were identified. We analysed the data of these patients from the time they first developed TB symptoms until they completed TB treatment. Our study adhered to the Strengthening The Reporting of Observational Studies in Epidemiology (STROBE) guidelines (see S1 Table in S1 File).

**Study setting.** The fieldwork for this study was conducted in Victoria by the VTP staff. Funded by the Victorian Government Department of Health, VTP is a centralised program located in metropolitan Victoria and works in partnership with hospitals and clinics in managing tuberculosis. All tuberculosis patients in Victoria are supervised by VTP nurse consultants [13, 14].

Victoria is the second most highly populated state in Australia, with 6.69 million residents as of March 2020 [15]. It has the highest population growth rate (1.8%), with net overseas migration functioning as the primary contributor to population growth in Victoria [15].

Victoria is divided into two distinct socio-geographic areas, metropolitan and regional.

Metropolitan is defined as the 31 local government areas of the city of Melbourne, while the 48 local government areas outside of the city are defined as regional. Based on 2016 census data, 4,485,211 of Victoria's total population of 5,926,624 lived within Melbourne, with less than 25% of the population living regionally [16]. This equates to an average population density of 500 people per square kilometre in metropolitan Victoria compared to an average of 6 people per square kilometre in regional Victoria.

There are some differences in the provision of health care services between Victoria's metropolitan and regional areas. Unlike patients in most metropolitan areas, patients in regional Victoria may reside a considerable distance from hospitals. Most regional hospitals do not have negative pressure rooms for isolating TB patients during their infectious stage; therefore, infectious patients need to be transferred to metropolitan hospitals. Some regional hospitals have no on-site TB specialist medical practitioners, and because of the distance from the hospital, there are very few opportunities for home visits from Victorian Tuberculosis Program specialist nursing staff, so regional patients must rely on telephone or video consultations for their diagnostic and follow-up consultations as well as their treatment supervision visits. The Victorian Tuberculosis Program (VTP) is Victoria's state-wide provider and coordinator of tuberculosis control.

In regional Victoria, the number of people born overseas is increasing. For example, in the 2006 census, there were 1,964 Indian-born people recorded as living in regional Victoria, which grew to 8,592 persons in 2016; similarly, the Philippines-born population was 2,700 in 2006 and 6,085 in 2016 [16]. The Australian Government has made changes to Commonwealth immigration policy intended to stimulate economic growth outside metropolitan areas in recent years. These various changes are focused on attracting migrants and international students to regional areas [17]. Historically, most migrants to Victoria have settled in the capital city of Melbourne, with many coming from countries with high TB incidence, such as India, the Philippines, and Sudan [18, 19]. Such changes to policy influence migration patterns

and may impact the distribution of TB cases within Victoria, which may also have implications for optimising health service delivery models [19].

## Study population

The population for this study included all people of all ages who had been diagnosed with tuberculosis and notified to the department of health.

**Inclusion criteria.** Patients were included in the study if they met the following inclusion criteria:

1. Diagnosed with TB in Victoria and notified to the Victorian department of health from 1 January 1995 to 31 December 2019. TB cases were defined in accordance with a standard national case definition based on either laboratory definitive evidence requiring isolation of Mycobacterium tuberculosis complex by culture or nucleic acid testing or clinical diagnosis accompanied by treatment [20].

2. Having received tuberculosis treatment in Victoria.

**Exclusion criteria.** Patients were excluded from the study if they:

1. were notified before 1995 or after 2019

2. Lacking residential addresses

**Variables.** The variables in this study included patient outcomes and predictors of the outcomes. The study outcomes were dependable variables: treatment outcomes and treatment delays. Treatment outcomes included completed treatment, lost to follow-up, died of TB, died of other causes during treatment for TB, and being transferred interstate or overseas.

Patients were defined as having died of TB when the clinical mode of death, the severity of TB disease (based on the presence of systemic symptoms, the extent of involvement of affected organs and central nervous system involvement), the presence of massive haemoptysis or respiratory, multisystem, or specific vital organ failure could be linked to tuberculosis and no other likely cause [21]. Died from other causes was defined as a death that was not attributed to tuberculosis based on pathological or autopsy examinations and death certificates indicating that the cause of death was another disease (21). Patients were defined as having completed treatment if they had completed a minimum of 6 months of therapy and assessed as having completed treatment by the treating physician [22]. Lost to follow-up was a treatment outcome for patients who did not complete therapy because they could not be located [23]. Because of a change of residency, some patients were transferred interstate or overseas to continue their treatment.

For the treatment delays, we adapted the definitions outlined by Van Wyk et al. [24], which proposed that: 'Patient treatment delay' is the period (in the number of days) between the onset of any self-reported TB symptoms and the first visit to a health care facility. 'Health system delay' is the period (in the number of days) between the first healthcare facility visit and initiation of TB treatment. 'Diagnostic delay' is defined as the period (in the number of days) between the onset of any self-reported TB-related symptoms and the time a chest x-ray was performed. 'Treatment initiation delay' was the period between a positive specimen (TB confirmed) and treatment initiation.

We extracted the following independent variables: (1) demographic data: age (age groups in years), sex (male or female), country of birth (name of the country), Aboriginal and Torres

Strait Islander status (Aboriginal and/or Torres Strait Islander or not Aboriginal and/or Torres Strait Islander), local government areas (local government area), self-reported residency status (Australian-born, permanent resident, refugee/humanitarian, visitor, overseas student, other and unknown status), and for overseas-born cases, year of arrival in Australia (year). (2) Clinical characteristics: year of tuberculosis notification (year), the manifestation of tuberculosis (pulmonary, extrapulmonary or both), chest X-ray results (abnormal, cavitation or normal), laboratory results (smear, culture, or gene expert). Most of these variables have been reported as influencing patient outcomes in some studies [22, 25].

## Ethical considerations

Approval from a Human Research Ethics Committee for this study was not required as the data were collected for the purposes of public health action, as defined in the Public Health and Wellbeing Act 2008 and were considered as being for quality assurance and auditing purposes. Patients were informed of the purpose of data collection and consented to their data being used for tuberculosis surveillance and medical research at the time of collection. All data were fully anonymised during the data extraction process. For example, participants' names, phone numbers, addresses, and birth dates were removed. Patient identification numbers, postcodes, and gender were coded. Age was changed to age group. In the publications that come from this study, patients will remain anonymous.

## Data analysis

Data cleaning and analyses were conducted using STATA version 14.

**Managing missing data.** We used a listwise deletion method when missing data contained residential addresses (participants were allocated neither to regional nor metropolitan areas) because our exposure of interest was regionality. When missing data did not have the key variable (i.e., residential address), we utilised the pairwise deletion approach, which allowed us to retain data and reduce the possibility of selection bias.

**Analysis.** Descriptive and multivariable analyses were performed. Incidence rates were calculated using the mid-year estimated resident population. We compared the number of TB cases and the TB incidence rates in regional and metropolitan areas from 1 January 1995 to 31 December 2019. Trends over this period were displayed graphically. Data were considered in 5 year increments, with the total number of notifications per 5 year category serving as the denominator. Given the recognised connection between age and risk of tuberculosis, we also stratified case notification by age for examination of overall trends.

We chose a more granular age category in Table 1 to describe the patients to determine which age group is impacted more by tuberculosis in order to plan for the targeted TB management program. The age groups < 65 years and ≥ 65 years were used in the univariate and multivariate analysis because, in some studies, the age groups ≥ 65 years have been found to be associated with poor treatment outcomes [25, 26].

A two-tailed p-value of <0.05 was considered statistically significant. In logistic regression, we compared complete treatment with death, irrespective of the cause, lost to follow-up and transferred interstate or overseas. Died of TB was compared with completed treatment, lost to follow-up, died of other causes during treatment for TB, and transferred interstate or overseas.

Area of residence, age, sex, drug susceptibility, and country of birth were included in univariable as well as multivariable analyses because there were found to affect TB treatment outcomes in some studies [22, 25, 27]. Our variable of interest was regionality. The proportional-hazards assumption was assessed using Kaplan-Meier survival curves by including time-dependent covariates in the model and with Schoenfeld residuals. In cases where

**Table 1. Characteristics of the notified cases of tuberculosis in Victoria from 1995 to 2019.**

| Variable | | Regional (n = 601) | | | | Metropolitan (n = 8,163) | | | |
|---|---|---|---|---|---|---|---|---|---|
| | | Overseas-born (n = 401) | | Australian-born (n = 200) | | Overseas-born (n = 7,375) | | Australian born (n = 788) | |
| | | Total | Proportion % | Total | Proportion % | Total | Proportion % | Total | Proportion % |
| Gender | Male | 215 | 54 | 123 | 62 | 3,869 | 52 | 447 | 57 |
| | Female | 186 | 46 | 77 | 39 | 3,500 | 47 | 339 | 43 |
| | Unknown | 0 | 0 | 0 | 0 | 6 | 0 | 2 | 0 |
| Age group | Under 5 | 3 | 1 | 11 | 6 | 31 | 0 | 115 | 15 |
| | 5–14 | 12 | 3 | 7 | 4 | 122 | 2 | 79 | 10 |
| | 15–24 | 49 | 12 | 14 | 7 | 1,303 | 18 | 129 | 16 |
| | 25–34 | 110 | 27 | 15 | 8 | 2,272 | 31 | 87 | 11 |
| | 35–44 | 60 | 15 | 10 | 5 | 1,188 | 16 | 60 | 8 |
| | 45–54 | 49 | 12 | 15 | 8 | 715 | 10 | 67 | 9 |
| | 55–64 | 33 | 8 | 31 | 16 | 562 | 8 | 56 | 7 |
| | 65 and above | 85 | 21 | 97 | 49 | 1,181 | 16 | 195 | 25 |
| | Unknown | 0 | 0 | 0 | 0 | 1 | 0 | 0 | 0 |
| Manifestation | Pulmonary | 193 | 48 | 131 | 66 | 2,864 | 39 | 459 | 58 |
| | Pulmonary Plus other sites | 42 | 10 | 18 | 9 | 936 | 13 | 117 | 15 |
| | Extrapulmonary | 162 | 40 | 49 | 25 | 3,396 | 46 | 206 | 26 |
| | Unknown | 4 | 1 | 2 | 1 | 179 | 2 | 6 | 1 |
| Susceptibility | Fully sensitive | 233 | 58 | 117 | 59 | 4,376 | 59 | 402 | 51 |
| | Multidrug -resistant tuberculosis | 4 | 1 | 0 | 0 | 88 | 1 | 9 | 1 |
| | Other resistance | 12 | 3 | 2 | 1 | 379 | 5 | 29 | 4 |
| | Extensively drug-resistant tuberculosis | 0 | 0 | 0 | 0 | 2 | 0 | 0 | 0 |
| | Genotypic Rifampicin resistant tuberculosis | 0 | 0 | 0 | 0 | 3 | 0 | 0 | 0 |
| | Unknown | 152 | 38 | 81 | 41 | 2,527 | 34 | 348 | 44 |
| Treat outcome (from 2005 to 2019 | Completed treatment | 238 | 86 | 88 | 82 | 4,491 | 90 | 442 | 92 |
| | Lost to follow-up | 5 | 2 | 4 | 4 | 107 | 2 | 8 | 2 |
| | Died from other cause | 12 | 4 | 12 | 11 | 128 | 3 | 22 | 5 |
| | Died of tuberculosis | 4 | 1 | 3 | 3 | 58 | 1 | 5 | 1 |
| | Transferred interstate or overseas | 18 | 6 | 0 | 0 | 222 | 4 | 1 | 0 |
| | Unknown | 1 | 0 | 0 | 0 | 1 | 0 | 0 | 0 |

Note: Treatment outcome regional, n = 385 and Metro, n = 5,485.

proportionality assumptions were not met, analyses were stratified. Kaplan-Meier survival curves were used to show various delays in presentation, diagnosis, and treatment between regional and metropolitan cohorts, and Cox proportional hazard analyses were performed to assess these delays.

Because of limited previous data, treatment outcomes were analysed using data from 2005 to 2019, and for the analysis of treatment delays, we used data from 2012 to 2019.

We described the follow-up period as follows: The time follow-up for patient delay started at symptom onset date, and the event was, the presentation to the healthcare centre. For healthcare system delay, follow-up time started at the presentation to the healthcare centre, and the event was, commenced on TB treatment. Diagnostic delay: follow-up time began at the presentation to the healthcare centre, and the event was, the first chest x-ray examination. Treatment delay: follow-up time started at first abnormal chest X-ray examination, and the

event was, commenced on TB treatment. Treatment delay 2: follow-up began on the first date the patient had a positive specimen for TB, and the event, was commenced on TB treatment. Censoring occurred when patients failed to have an event because they either transferred inter-state/overseas or dies or lost to follow-up (right censor).

## Results

A total of 8,819 TB cases were notified to the Victorian Government Department of Health between 1995 and 2019. Among the 8,819 cases, 611 (7%) were recorded in regional areas, 8,163 (93%) in metropolitan areas of Victoria and 45 (1%) had neither regional nor metropolitan residential addresses (see Fig 1). Forty-five cases with no residential addresses were excluded from the study as they were classified as neither regional nor metropolitan. Of the 611 people in regional areas, 343 (56%) were male, 401 (66%) were overseas-born, and for 10 cases (2%), there was no country of birth recorded. Among the 8,163 TB cases in metropolitan areas, 4,316 (53%) were male, 8 (0.1%) had no gender reported and 7,375 (90%) were overseas-born.

Data recorded before 2005 had missing treatment outcomes for many cases and were therefore excluded from the analysis of treatment outcomes (Fig 1). The overall treatment completion rates were similar among the regional and metropolitan cohorts: 85% and 90%, respectively.

Table 1 describes the characteristics of notified TB cases in Victoria from 1995 to 2019 by location of residence and birth. The proportion of overseas-born cases in the regional cohort was 66% compared to 90%, in metropolitan areas. The 25 to 34 age group had the largest proportion of cases in both regional and metropolitan areas. In this age bracket, there were 27% of overseas-born and 8% of Australian-born cases in regional settings, and 31% of overseas-born and 11% of Australian born cases in metropolitan areas. The proportions of multidrug-resistant TB (MDR-TB) cases among the regional and metropolitan patients were similar. The four MDR-TB cases reported in the regional cohort all occurred in people born overseas. In the metropolitan cohort, the proportion of MDR-TB cases was the same (1%) amongst overseas and Australian-born persons. Extensive drug-resistant TB (0.03%) and genotypic rifampicin-resistant TB (0.04%) were only reported in the metropolitan area among overseas-born cases.

The number of tuberculosis cases in regional Victoria has fluctuated over time, with 129 notified from 1995 to 1999, 97 from 2000 to 2004, and 155 from 2015 to 2019. Table 2 compares the TB incidence rate between regional and metropolitan areas. From 1995 to 1999, there were 129 cases with a mean incidence rate of 2.0, 95% CI 1.3–2.7 per 100,000 population in regional Victoria, while in the metropolitan there were 1,271 cases with a mean incidence rate of 7.7, 95% CI 6.9–8.4. The TB incidence in regional and metropolitan areas is fluctuating; in the 2015 to 2019 period, the mean incidence for the regional cohort was 2.1, 95% CI 1.5–2.7 and 7.9, 95% CI 7.3–8.4 in the metropolitan.

In the overseas-born cohort, 29 people were under the age of 20, 236 were between the ages of 20 and 49, and 136 were ≥50 (Fig 2). Twenty cases were under the age of 20, 43 were aged 20–49 years, and 137 were ≥50 in the Australian-born cohort. TB cases among overseas-born people aged 20–49 years increased from 22 to 79 cases between 2000 and 2019. Conversely, Australian-born cases in this age bracket remained effectively unchanged over the same period (11 cases vs 10 cases).

A total of 386 overseas-born cases had their year of arrival in Australia recorded. Half of the people (197; 51%) developed TB disease within five years of arrival in Australia. Out of these 197 cases, 58 (29%) were permanent residents, 43 (22%) were refugees, 19 (10%) were visitors, 18 (9%) were overseas students, and 59 (30%) had unknown residential status. Among 19

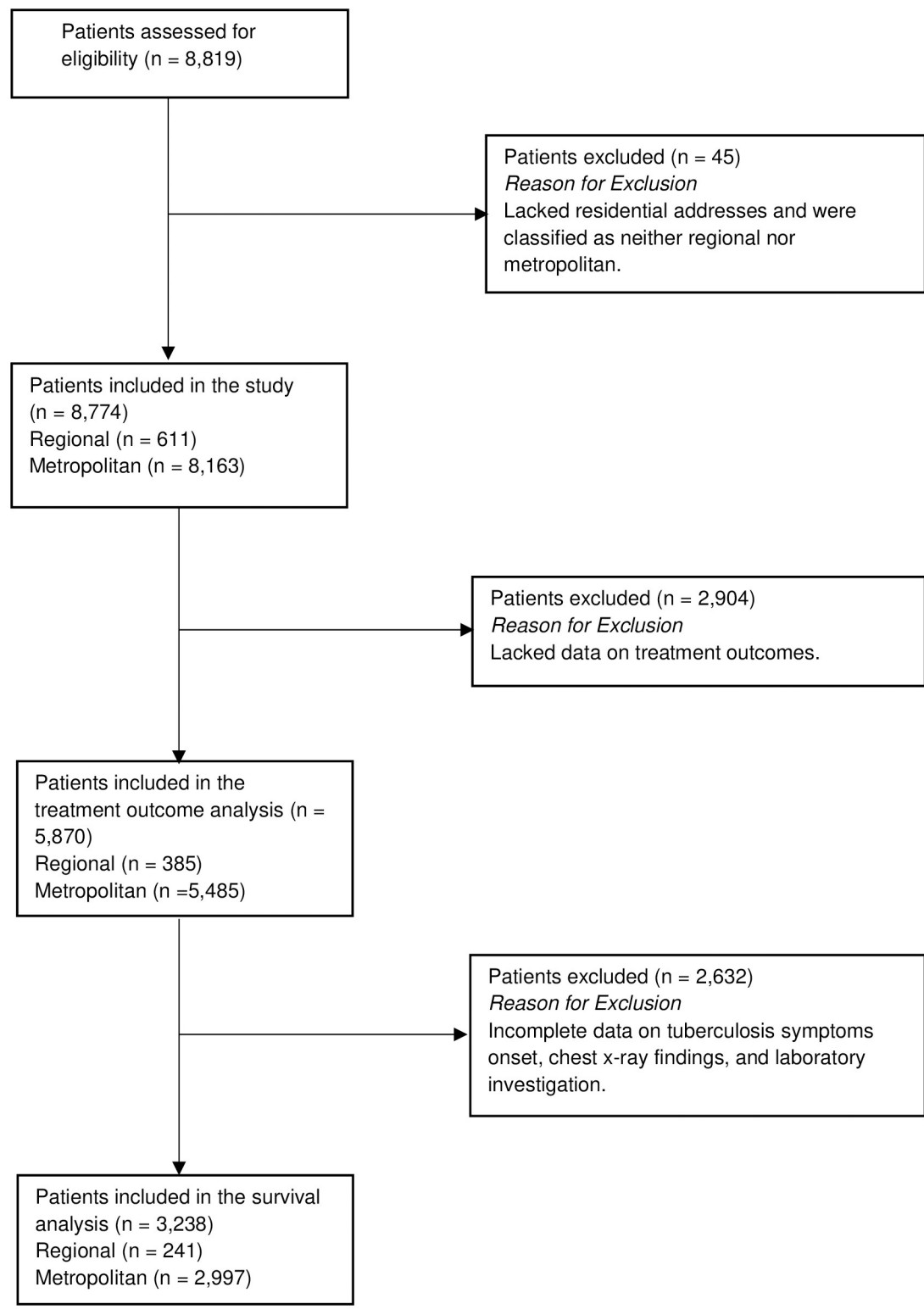

**Fig 1. The flow of patients through the study.**

**Table 2. Tuberculosis incidence rate per 100,000 population in regional Victoria and metropolitan Victoria from 1995 to 2019.**

| Years | Regional areas | | Metropolitan areas | |
|---|---|---|---|---|
| | Tuberculosis cases | Mean incidence rate per 100,000 population (95% CI) | Tuberculosis cases | Mean incidence rate per 100,000 population (95% CI) |
| 1995–1999 | 129 | 2.0 (1.3–2.7) | 1,271 | 7.7 (6.9–8.4) |
| 2000–2004 | 97 | 1.5 (1.0–1.9) | 1,407 | 8.0 (7.6–8.4) |
| 2005–2009 | 98 | 1.4 (1.2–1.6) | 1,742 | 9.1 (8.9–9.4) |
| 2010–2014 | 132 | 1.8 (1.4–2.1) | 1,841 | 8.7 (7.9–9.5) |
| 2015–2019 | 155 | 2.1 (1.5–2.7) | 1,902 | 7.9 (7.3–8.4) |

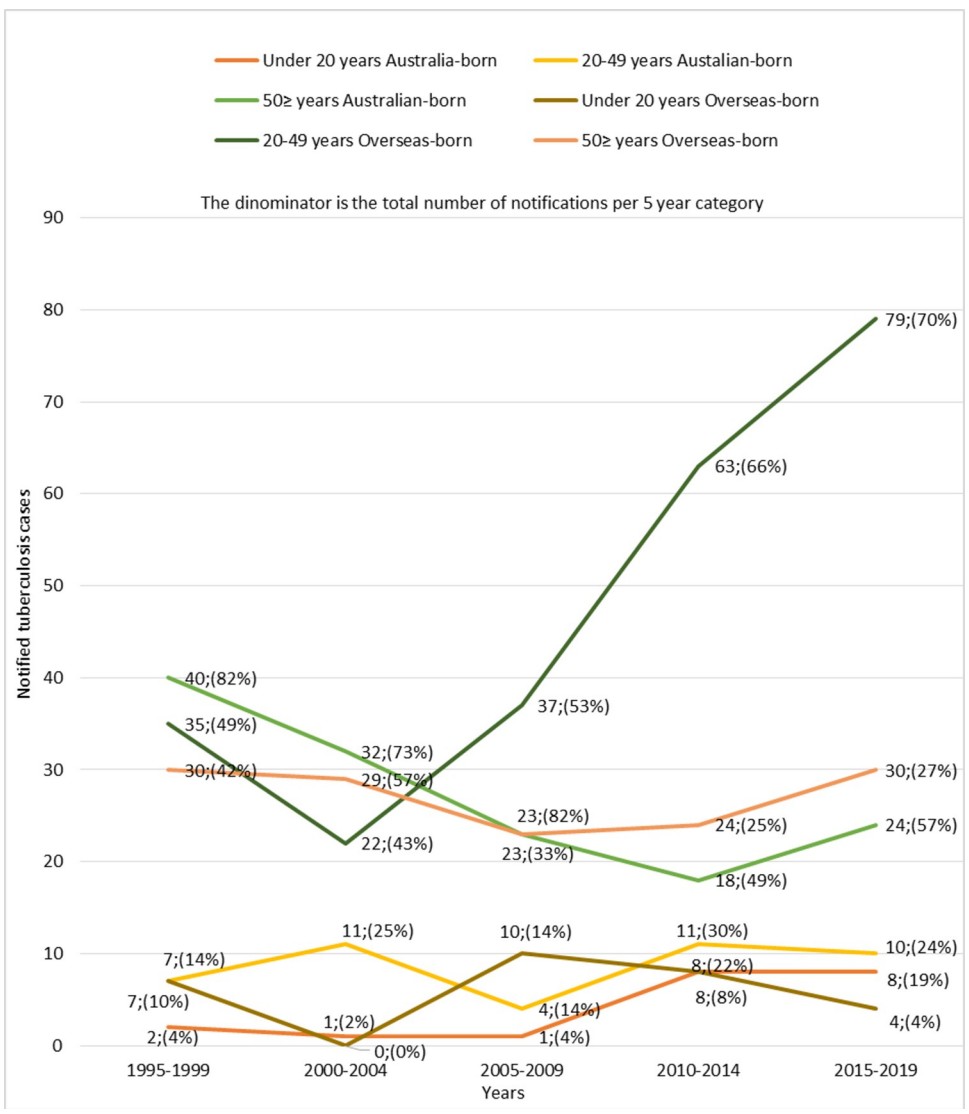

**Fig 2. Tuberculosis cases in regional Victoria by year of notification, age, and country of birth from 1995 to 2019.**

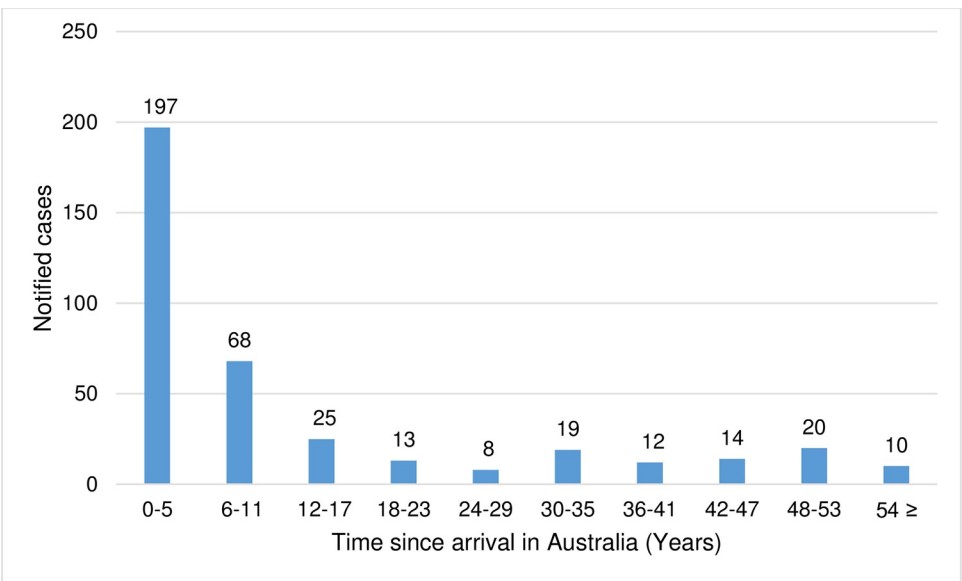

**Fig 3. Notified cases of tuberculosis in the overseas-born people in regional Victoria from 1995–2019 by the number of years since arrival in Australia.**

overseas students, 18 developed TB within five years and one between six and 11 years of arrival. Of the 46 refugees recorded in the study, 43 were diagnosed with TB within five years of arrival. The risk of developing TB remained for many years after people arrived in Australia; 10 (3%) people were diagnosed with TB after 53 years of arrival (see, Fig 3).

In 2016, the TB incidence rate for the Australian-born population was 0.5 per 100,000 people and 12.0 per 100,000 people born overseas. Thirteen frequently reported countries of birth for the overseas-born cases from 1999 to 2019 are shown in Fig 4. People born in these 13 countries make up 62% of the overseas-born cases during the study period. India had the highest number of notified cases, 58 (14%), followed by the Philippines, 48 (12%). People born in India and the Philippines accounted for 26% of all the TB cases in regional areas.

Among 5,870 cases with known treatment outcomes, 5,259 (90%) completed treatment, 124 (2%) were lost to follow-up, 174 (3%) died of another cause while on TB therapy, 70 (1%) died of TB, 2 (0.03%) were still on treatment at the time of data extraction, and 241 (4%) were transferred either interstate or overseas. Table 3 shows univariable and multivariable analyses of predictors of treatment completion and dying of TB. Living in a regional area was associated with lower odds of treatment completion on univariable analysis (OR = 0.6, 95% CI 0.5–0.8, P = 0.002). After adjusting for the effect of age, sex, drug susceptibility and country of birth in the model, living in a regional area remained significantly associated with lower treatment completion (Adjusted OR [AOR] = 0.7, 95% CI 0.5–0.9, P = 0.019). On multivariable analysis, older age and male sex were also predictors of lower treatment completion.

Living in a regional area did not significantly increase the odds of dying from TB on univariable analysis (OR = 1.6, 95% CI 0.7–3.5, P = 0.244). In a model that included all the five variables in the multivariable analysis, regionality was not associated with dying of TB (AOR = 1.8, 95% CI 0.7–4.2, P = 0.198). Older age was significantly associated with dying of TB.

Data prior to 2012 were incomplete in relation to TB symptom onset, chest X-ray findings and laboratory investigations and were therefore unable to be included in the analyses of delay in TB diagnosis and treatment. A total of 241 cases in regional and 2,997 in metropolitan areas

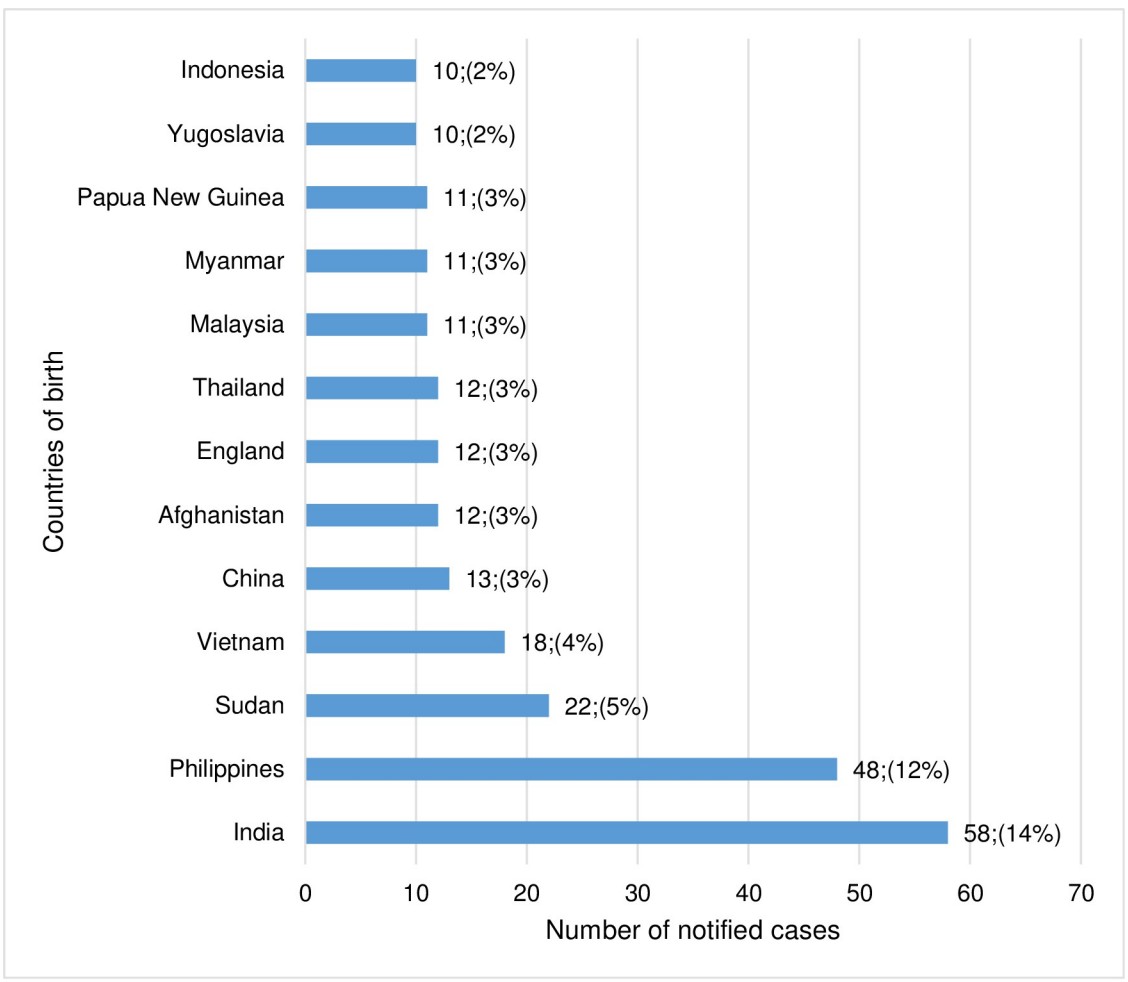

Notes: Yugoslavia no longer exist.

**Fig 4. Notified cases of tuberculosis in regional Victoria for overseas-born people from 1999 to 2019, by country of birth.**

were included. Table 4 shows the adjusted relationship between regionality in Cox regression models of delays in the cascade of care among tuberculosis patients in Victoria from 2012 to 2019. Patient and health system delays were similar in regional and metropolitan areas for cases with pulmonary involvement. In regional areas, people with pulmonary involvement underwent chest x-ray (diagnostic delay) slightly sooner than those notified in metropolitan areas (median six days versus nine days, AHR = 1.2, 95% CI 1.0–1.5, P = 0.047). Conversely, healthcare system delay was non-significantly longer in regional than in metropolitan patients (median 64 days versus 54 days, AHR = 0.8, 95% CI 0.6–1.0, P = 0.094).

The Kaplan-Meier curves for patient delay are shown in Fig 5.

Table 5 shows patients who did not have an event in the survival analysis. Among 1,343 patients with pulmonary TB in the metropolitan area, 161 did not undergo chest x-ray examination after their first presentation to the health care centre compared to 16 out of 122 in the rural area. Nine out of 1,107 people in metropolitan with extrapulmonary TB did not commence treatment after their first presentation to the healthcare centre compared to none in rural areas. The main reasons for not having an event in the treatment delay analysis were lost

**Table 3. Univariable and multivariable analysis of predictors of treatment completion and dying of tuberculosis from 2005 to 2019.**

| Variable | Treatment completion | | | | Died of tuberculosis | | | |
|---|---|---|---|---|---|---|---|---|
| | Univariable analysis | | Multivariable analysis | | Univariable analysis | | Multivariable analysis | |
| | OR (95%) | P valve | AOR (95%) | P valve | OR (95%) | P valve | AOR (95%) | P valve |
| Regional (reference group: metropolitan cases) | 0.6 (0.5–0.8) | 0.002 | 0.7 (0.5–0.9) | 0.019 | 1.6 (0.7–3.5) | 0.244 | 1.8 (0.7–4.2) | 0.198 |
| Age ≥ 65 years (reference group: age < 65 years) | 0.2 (0.2–0.3) | <0.0001 | 0.2 (0.2–0.3) | <0.0001 | 12.6 (7.6–20.9) | <0.0001 | 9.9 (5.8–16.9) | <0.0001 |
| Male sex (reference group: female sex) | 0.6 (0.5–0.7) | <0.0001 | 0.7 (0.6–0.8) | <0.0001 | 1.8 (1.1–3.0) | 0.020 | 1.5 (0.9–2.6) | 0.131 |
| Drug resistant tuberculosis (reference group: fully sensitive tuberculosis) | 0.8 (0.6–1.1) | 0.228 | 0.8 (0.6–1.1) | 0.128 | 1.7 (0.8–3.5) | 0.179 | 1.9 (0.9–4.1) | 0.104 |
| Overseas-born (reference group: Australian-born | 0.9 (0.7–1.2) | 0.415 | 0.7 (0.5–1.1) | 0.094 | 0.9 (0.4–1.8) | 0.682 | 1.8 (0.7–4.9) | 0.238 |

Notes: AOR is an adjusted odds ratio.

to follow-up, died of another cause, or died of TB, or transferred interstate or overseas before the event. For example, in health system delay for patients with pulmonary involvement, in the rural cohort, one patient died from another cause, two died of TB and one transferred interstate or overseas after first presenting to the health care centre but before commencing TB therapy. In the metropolitan cohort, one was lost to follow-up, eight died from another cause, 12 died of TB, and 17 transferred interstate or overseas before commencing treatment.

## Discussion

We report the trends and treatment outcomes of notified TB cases in regional areas of Victoria, Australia, from 1995 to 2019. The incidence of TB is low in regional areas, and this is

**Table 4. Adjusted relationship between regionality in Cox regression models of delays in the cascade of care among tuberculosis patients in Victoria from 2012 to 2019.**

| Time period outcome | Regional area Median (interquartile range) days | Metropolitan area Median (interquartile range) days | Effect of regional in Cox regression analysis | | | |
|---|---|---|---|---|---|---|
| | | | Number observed in regional | Number observed in metropolitan | Adjusted Hazard ratio (95% CI) | P-value |
| Patient with pulmonary involvement | | | | | | |
| Patient delay | 21 (1–76) | 24 (1–67) | 113 | 1,293 | 0.9 (0.8–1.1) | 0.389 |
| Health system delay | 21.5(7–44) | 25 (7–65) | 146 | 1,691 | 1.2 (1.0–1.4) | 0.102 |
| Diagnostic delay: Presentation to First chest-x-ray | 6 (0–27) | 9 (0–41) | 122 | 1,343 | 1.2 (1.0–1.5) | 0.047 |
| Treatment delay: First chest X-ray to start of tuberculosis treatment | 8 (3–23) | 9 (3–29) | 131 | 1,565 | 1.0 (0.9–1.3) | 0.614 |
| Extrapulmonary patients | | | | | | |
| Patient delay | 11.5 (0–68) | 23 (0–75) | 62 | 907 | 1.0 (0.7–1.2) | 0.756 |
| Health system delay | 64 (26–137) | 54 (21–112) | 78 | 1,107 | 0.8 (0.6–1.0) | 0.094 |
| Diagnostic delay: Presentation to First chest x-ray | 26 (5–92) | 30 (5–72) | 47 | 798 | 0.9 (0.7–1.2) | 0.393 |
| All patients | | | | | | |
| Treatment delay 2: specimen test to treatment initiation | 7 (1–19) | 5 (1–17) | 151 | 1,996 | 1.0 (0.8–1.2) | 0.747 |

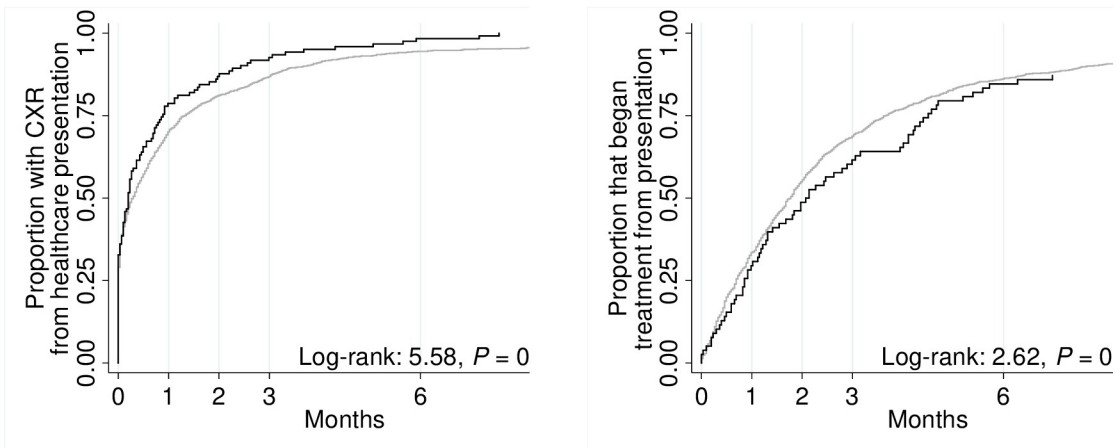

**Fig 5. Kaplan-Meier curves in tuberculosis patients in Victoria, Australia from 2012 to 2019.**

consistent with the findings from studies reported in the United States of America (USA) and the United Kingdom. For example, a study in Appalachia, USA, reported that in 2005 the rate of tuberculosis in regional Appalachia was 2.1 compared to 2.7 per 100,000 population in metropolitan areas [8]. A similar study conducted in England and Wales from 2001 to 2003 found that the rate of TB in metropolitan areas was 6.3-fold higher compared to regional areas [9].

**Table 5. Patients who did not have an event in the survival analysis.**

| | Patients who did have an event | Patients who did not have an event | Patients who did have an event | Patients who did not have an event |
|---|---|---|---|---|
| Patients with pulmonary involvement | Rural | | Metropolitan | |
| Patient delay. Event: attending health centre after experiencing TB symptoms | 113 | 0 | 1,293 | 0 |
| Health system delay. Event: starting treatment after first health care centre visit | 146 | 4 | 1,691 | 38 |
| Diagnosis delay. Event: having chest x-ray after the first presentation to the health care centre | 122 | 16 | 1,343 | 161 |
| Treatment delay: Event: starting tuberculosis treatment after undergoing first chest x-ray | 131 | 3 | 1,565 | 21 |
| Extrapulmonary patients | | | | |
| Patient delay. Event: attending health centre after experiencing TB symptoms | 62 | 0 | 907 | 0 |
| Health system delay. Event: starting treatment after first health care centre visit | 78 | 0 | 1,107 | 9 |
| Diagnosis delay. Event: having chest x-ray after the first presentation to the health care centre | 47 | 22 | 798 | 218 |
| Treatment delay 2. Event: treatment initiation after first positive TB specimen test | 151 | 7 | 1,996 | 51 |

We observed that TB incidence in regional Victoria from 1995 to 2019 has remained essentially stable, but that there was an increase in TB cases aged between 20 and 49 years among the overseas-born in the regional cohort. The observed increase in TB incidence is group is consistent with changing demographics reflecting migration of working-age people from high TB incidence countries [16]. In the regional cohort, the proportion of TB among the overseas-born population was twice that of the Australia-born people. These results suggest there may be a public health benefit in increasing access to testing services, including latent TB detection and treatment in regional areas, targeting 20-49-year-old overseas-born people, and offering TB preventative therapy to those found with latent TB.

In regional Victoria, TB resistance was more common in overseas-born cases, consistent with other Australian studies [10, 28]. We analysed the time from arrival in Australia to TB diagnosis for overseas-born cases in the regional cohort. More than half of the overseas-born cases were notified within five years of arrival in Australia. The high TB notification within the first five years of arrival may be attributed to the latent TB reactivation [29]. It is worth noting that the majority of the refugees were diagnosed within five years of arrival. This may be a result of more intensive screening soon after arrival, including testing for LTBI in asylum seekers but not migrants more generally. Refugees may also return overseas less frequently than other migrants and be less likely to be reinfected.

In this study, health-seeking behaviour and treatment outcomes were similar between the regional and metropolitan settings. These results suggest that existing programs are functioning well, although the possible trend toward health service delays requires further monitoring and reviewing opportunities for programmatic strengthening. In addition, people aged over 64 years are at significantly greater risk of dying from TB, and in appreciation of this risk, more intensive care may be required. Our analysis indicates that regionality is not an independent determinant of dying of TB. This is inconsistent with a study by Mutembo et al. [30], which reported that rural locations in Zambia had a 70% higher risk of death; however, these differences have occurred in the context of our higher resourced and lower incidence setting. We did identify that older age and male sex were predictors of lower treatment completion in regional Victoria, which may allow for interventions in our context. In particular, awareness of lower rates of treatment completion in these groups can be considered in assessing the level of follow-up support and medication supervision required for people with TB and encourage targeted education and adherence support services to groups with observed lower rates of completion.

Due to a paucity of research in regional areas of countries with a low incidence of TB, we cannot make a direct comparison between our study and other published literature. Putting aside the comparison between regional and metropolitan data, our results in relation to delays in TB diagnosis and treatment are consistent with other Australian studies as well as systematic reviews [7, 14, 31]. For example, Bello et al. [31] performed a systematic review of 198 studies. They reported a median duration of patient delay of 28 days and a health system delay of 18 days compared to 21 days for each of these categories for the regional patients in our study. Of interest, extrapulmonary tuberculosis in our cohort had a much longer health system delay, averaging 64 days for regional patients.

## Limitations

Strengths of this study include the use of a comprehensive central database that includes important demographic, clinical and laboratory data, allowing for the incorporation of other factors outlined in this manuscript and a long study period of 25 years. However, we acknowledge that data for the entire study period are not available for all data fields (e.g., treatment

outcomes, health system delays), limiting trend analysis. Some of the data in our study, such as dates of symptom onset and healthcare presentation, were collected retrospectively from patients and thus may contain inaccuracies relating to recall bias. Data on some factors that could have influenced the treatment delay and outcomes, such as educational level was limited.

## Conclusion

Tuberculosis in regional Victoria is more common among the overseas-born population, and patients with extrapulmonary TB in regional areas have non-significant minor delays in treatment commencement. Increasing migration from high incidence TB countries to regional settings in Australia requires an ongoing review of available and accessible health services to limit delays in timely diagnosis and treatment. Increasing access to LTBI management and enhanced diagnostic pathways in regional areas may assist in reducing the burden and impact of TB in the future.

## Supporting information

**S1 File.**
(DOCX)

## Acknowledgments

The authors would like to thank the clinical nurse consultants for collecting and entering data into the database. We want to thank Peta Edler for her advice on our data analysis.

## Author Contributions

**Conceptualization:** Nompilo Moyo, Justin T. Denholm.

**Data curation:** Ee Laine Tay.

**Formal analysis:** Nompilo Moyo, Ee Laine Tay.

**Investigation:** Nompilo Moyo.

**Methodology:** Nompilo Moyo, Justin T. Denholm.

**Project administration:** Nompilo Moyo.

**Supervision:** Justin T. Denholm.

**Writing – original draft:** Nompilo Moyo.

**Writing – review & editing:** Nompilo Moyo, Ee Laine Tay, James M. Trauer, Leona Burke, Justin Jackson, Robert J. Commons, Sarah C. Boyd, Kasha P. Singh, Justin T. Denholm.

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
