## [Decision Letter · Decision Letter 0]

11 Mar 2022

PONE-D-22-00893Tuberculosis notifications in regional Victoria, Australia: implications for public health care in a low incidence settingPLOS ONE

Dear Dr. Moyo,

Thank you for submitting your manuscript to PLOS ONE. After careful consideration, we feel that it has merit but does not fully meet PLOS ONE’s publication criteria as it currently stands. Therefore, we invite you to submit a revised version of the manuscript that addresses the points raised during the review process.

We look forward to receiving your revised manuscript.

Kind regards,

Mohamed F. Jalloh, PhD, MPH

Academic Editor

PLOS ONE

Journal Requirements:

Reviewers' comments:

Reviewer's Responses to Questions

**Comments to the Author**

1. Is the manuscript technically sound, and do the data support the conclusions?

Reviewer #1: Partly

Reviewer #2: Yes

2. Has the statistical analysis been performed appropriately and rigorously? 

Reviewer #1: Yes

Reviewer #2: Yes

3. Have the authors made all data underlying the findings in their manuscript fully available?

Reviewer #1: Yes

Reviewer #2: Yes

4. Is the manuscript presented in an intelligible fashion and written in standard English?

Reviewer #1: Yes

Reviewer #2: No

5. Review Comments to the Author

Reviewer #1: The analyses are relatively straightforward but cover an important topic related to an important infectious disease (TB). The findings are relevant to understanding dynamics of tuberculosis in Victoria, Australia as well as potentially in other parts of Australia and other countries with similar population dynamics and underlying epidemiological conditions. However, as currently submitted the presentation of some of the findings as well as the unacceptable preponderance of typographical and grammatical errors need to be addressed. Some more specific comments are included below:

Line 59 – Consider defining what “regional” means in the context up front. The implication is that it refers to non-metropolitan areas but that will not become completely lear to non-Australian audiences until lines 112-115

Line 66-67 – is there a reference to back up the statement that many migrants arrive from countries known to have a high TB incidence?

Line 72-74 – it seems as if more was done that to describe notified TB cases…

Line 128 – Ethical ConsiderationS

Line 131 – should be “were considered” since this refers to data

Line 135 – “and” missing

Line 159 – excessive “and”

Line 169-170 – the overseas cases proportions are already mentioned in Lines 161-3

Table 1 – formatting with some proportions rounded to whole numbers and other to one or two decimal points

Line 189-190 – it misrepresents the trend to say that there was a slight increase in TB notification over time. It is true when comparing the two time points but over time it actually fluctuated quite a bit including notable decreases over a few of the time periods (see Table 2)

Line 208-209 – median time to diagnosis AFTER ARRIVAL

Line 209-210 – In this particular case it is repetitive of the median cited in the prior sentence to say that half developed TB disease within 5 years (this is the Median)

Line 234-235 – Specify the directionality of the association (in this case an association between living in a regional area and lower odds of treatment completion).

Line 290 – reference for latent TB reactivation?

Figure 2 - Needs to be reworked entirely. For example, it is inappropriate to have stacked bars adding up to 200%. In addition, the data labels can be presented in a less crowded, more easily legible way.

Reviewer #2: Background

1. Line 59: Authors introduce the term “regional Victoria”.

Comment: I suggest authors add a description what this means as it will be helpful for readers who are not familiar with Australia.

2. Line 65-67: “Historically, most migrants to Victoria have been based in the capital city of Melbourne, with many arriving from countries known to have a high TB incidence”.

Comment: Authors should mention examples of these countries and add references.

3. Line 67-69: “Such changes to policy influence migration patterns and may impact the distribution of TB cases within Victoria”.

Comment: Please add a reference to support the claim of migration and TB cases.

Methods

1. Authors should consider improving this section by adding clear sub-sections, for example data source, study population, variables, statistical analysis etc.

2. Line 82-91: The description covered in line 82-91 seems to better fit elsewhere, either in the introduction or discussion sections and not in the methods section.

3. Line 100-106: The description of variables need to be improved. Indicate clearly the dependent and independent variables. Also, if variables are continuous or categorical and describe the categories. For example, sex (male or female), age (are there categories or continuous, age range?).

4. Description of the study population is missing, including inclusion or exclusion of observations and dealing with missing information.

Results

1. Line 160: description of exclusion of observations should be in the Methods section.

2. Line 164-167: again, description of exclusion of observations should be in the Methods section

3. Line 199-200: table 2, the numbers are not properly formatted. Please revise.

4. Line 243: Seems multivariable and multivariate are used interchangeably. Please maintain uniformity, use multivariable.

Discussion

1. Line 289: “The higher TB rate among the overseas-born people could be attributed to latent TB reactivation”

Comment: Authors need to add references to support this statement.

2. Authors should add a study limitation section

6. PLOS authors have the option to publish the peer review history of their article (what does this mean?). If published, this will include your full peer review and any attached files.

Reviewer #1: No

Reviewer #2: **Yes: **Alexander Kailembo

---

## [Author Response · Author response to Decision Letter 0]

13 Apr 2022

Reviewer 1 

1. Your feedback. The analyses are relatively straightforward but cover an important topic related to an important infectious disease (TB). The findings are relevant to understanding dynamics of tuberculosis in Victoria, Australia as well as potentially in other parts of Australia and other countries with similar population dynamics and underlying epidemiological conditions. However, as currently submitted the presentation of some of the findings as well as the unacceptable preponderance of typographical and grammatical errors need to be addressed. Some more specific comments are included below:

Our response: Thank you for these comments, which we have responded to in detail below.

2. Your feedback. Line 59 – Consider defining what “regional” means in the context up front. The implication is that it refers to non-metropolitan areas but that will not become completely lear to non-Australian audiences until lines 112-115.

Our response. We have moved the definition of regional into introductory material, and the statement now reads: “Victoria is divided into two distinct geographic areas: metropolitan and regional. Metropolitan is the geographical area that defines Melbourne as the city and capital of the state of Victoria and has 31 local government areas. Regional is defined as a geographical area outside Melbourne city and includes 48 (out of a total of 79) local government areas from across the state [2]. Based on 2016 census data, 4,485,211 of Victoria’s total population of 5,926,624 lived within Melbourne, with less than 25% of the population living regionally (3). This equates to an average population density of 500 people per square kilometre in metropolitan Victoria compared to an average of 6 people per square kilometre in regional Victoria.” Lines 59-67.

3. Your feedback. Line 66-67 – is there a reference to back up the statement that many migrants arrive from countries known to have a high TB incidence? 

Our response. We have added the reference and the statement now reads “Historically, most migrants to Victoria have settled in the capital city of Melbourne, with many coming from countries with high TB incidence, such as India, the Philippines, and Sudan (5, 6).” Lines 85-87.

4. Your feedback. Line 72-74 – it seems as if more was done that to describe notified TB cases… 

Our response. We have provided a description of notified TB cases by region in Table 1. It’s not clear to us from this comment that changes to our manuscript are requested, but if specific additional information would be helpful editorially, we would be happy to consider it.

5. Your feedback. Line 128 – Ethical ConsiderationS

Our response. We have corrected the text and it now reads “Ethical Considerations” (Line 156).

6. Your feedback. Line 131 – should be “were considered” since this refers to data. 

Our response. "Was considered" has been replaced with "were considered." Line 159.

7. Your feedback. Line 135 – “and” missing. 

Our response. The sentence was amended to “For example, names, phone numbers, addresses, and dates of birth of participants were removed.” Lines 162-163.

8. Your feedback. Line 159 – excessive “and” 

Our response. We amended the statement, and it now reads “Forty-five cases were excluded from the study because they lacked residential addresses and were classified as neither regional nor metropolitan.” Lines 195-197.

9. Your feedback. Line 169-170 – the overseas cases proportions are already mentioned in Lines 161-3

Our response. We reported two different proportions in these statements. The proportions reported in lines 161-3 were the overall proportions of the two cohorts, whereas the proportions described in lines 169-170 were specific to drug susceptibility. Lines 210-216.

10. Your feedback. Table 1 – formatting with some proportions rounded to whole numbers and other to one or two decimal points. 

Our response. We have rounded the numbers in table 1 to whole numbers.

11. Line 189-190 – it misrepresents the trend to say that there was a slight increase in TB notification over time. It is true when comparing the two time points but over time it actually fluctuated quite a bit including notable decreases over a few of the time periods (see Table 2)

Our response. We amended the text, and it now reads “The number of tuberculosis cases in regional Victoria has fluctuated over time, with 129 notified from 1995 to 1999, 97 from 2000 to 2004, and 155 from 2015 to 2019.” Lines 238-239. 

12. Your feedback. Line 208-209 – median time to diagnosis AFTER ARRIVAL. Line 209-210 – In this particular case it is repetitive of the median cited in the prior sentence to say that half developed TB disease within 5 years (this is the Median)

Our response. The statement “The median time to diagnosis was five years, and the interquartile range was 1 to 17 years” has been deleted. 

13. Line 234-235 – Specify the directionality of the association (in this case an association between living in a regional area and lower odds of treatment completion). 

Our response: We have amended the statement and it now reads “Living in a regional area was associated with lower odds of treatment completion on univariable analysis (OR = 0.6, 95% CI 0.5-0.8, P = 0.002).” Lines 283-285.

14. Your feedback. Line 290 – reference for latent TB reactivation? 

Our response: We have added a reference to support the statement. The high TB notification within the first five years of arrival may be attributed to latent TB reactivation (15). Lines 348-349.

15. Your feedback. Figure 2 - Needs to be reworked entirely. For example, it is inappropriate to have stacked bars adding up to 200%. In addition, the data labels can be presented in a less crowded, more easily legible way. 

Our response. Figure 2 has been amended. The residency status and the percentages in the data labels have been deleted. Figure 2 now shows the notified cases of tuberculosis in overseas-born people in regional Victoria from 1995-2019 by the number of years since arrival in Australia. 

Fig 2. Notified cases of tuberculosis in overseas-born people in regional Victoria from 1995-2019 by the number of years since arrival in Australia.

Reviewer 2

Background

1. Your feedback. Line 59: Authors introduce the term “regional Victoria”.

Comment: I suggest authors add a description what this means as it will be helpful for readers who are not familiar with Australia. 

Our response: We have amended the manuscript to reflect this feedback. The manuscript now reads “Victoria is divided into two distinct geographic areas: metropolitan and regional. Metropolitan is the geographical area that defines Melbourne as the city and capital of the state of Victoria and has 31 local government areas. Regional is defined as a geographical area outside Melbourne city and includes 48 (out of a total of 79) local government areas from across the state [2]." Lines 59-67.

2. Your feedback. Line 65-67: “Historically, most migrants to Victoria have been based in the capital city of Melbourne, with many arriving from countries known to have a high TB incidence”.

Comment: Authors should mention examples of these countries and add references. Philippines, India, and Vietnam.

Our response. We have added the countries and supported the statement with a reference. This now reads: “Historically, most migrants to Victoria have settled in the capital city of Melbourne, with many coming from countries with high TB incidence, such as India, the Philippines, and Sudan [5, 6].” Lines 85-87.

3. Your feedback. Line 67-69: “Such changes to policy influence migration patterns and may impact the distribution of TB cases within Victoria”.

Comment: Please add a reference to support the claim of migration and TB cases.

Our response. We have added the reference to support the statement. The manuscript now reads “Historically, most migrants to Victoria have settled in the capital city of Melbourne, with many coming from countries with high TB incidence, such as India, the Philippines, and Sudan [5,6].” Lines 85-87.

Methods

1. Your feedback. Authors should consider improving this section by adding clear sub-sections, for example data source, study population, variables, statistical analysis etc. 

Our response. Thank you for this suggestion. We have added these subsections to the methods section of the manuscript, most significantly reflected in lines 96-190.

2. Your feedback. Line 82-91: The description covered in line 82-91 seems to better fit elsewhere, either in the introduction or discussion sections and not in the methods section.

Our response. The description in line 82-91 has been moved to the Background section, and is now found in lines 69-78.

3. Your feedback. Line 100-106: The description of variables need to be improved. Indicate clearly the dependent and independent variables. Also, if variables are continuous or categorical and describe the categories. For example, sex (male or female), age (are there categories or continuous, age range?). 

Our feedback. We have amended the paragraph and it now reads “We extracted the following variables: (1) demographic data: age (age groups in years), sex (male or female), country of birth (name of the country), Aboriginal and/or Torres Strait Islander status (Aboriginal and/or Torres Strait Islander or not Aboriginal and/or Torres Strait Islander), local government areas (local government area), self-reported residency status (Australian-born, permanent resident, refugee/humanitarian, visitor, overseas student, other or unknown status), and for overseas-born cases, year of arrival in Australia (year). (2) Clinical characteristics: year of tuberculosis notification (year), the manifestation of tuberculosis (pulmonary, extrapulmonary or both), chest X-ray results (abnormal, cavitation or normal), laboratory results (smear, culture, or gene expert), and treatment outcome (died of TB , died of other causes during treatment for TB, completed treatment, lost to follow up, and transferred interstate or overseas). The independent variables include age, gender, country of birth, and place of residency. Treatment outcomes and treatment delays are the dependent variables.” This information is now presented in a new section, in lines 131-144.

4. Your feedback: Description of the study population is missing, including inclusion or exclusion of observations and dealing with missing information.

Our response. We added the following text to the methods section of the manuscript: “We extracted TB surveillance data. Patients were included in the study if they met the following inclusion criteria: (1) diagnosed with TB in Victoria and notified to the Victorian department of health from 1 January 1995 to 31 December 2019, with TB cases defined in accordance with a standard national case definition based on either laboratory definitive evidence requiring isolation of Mycobacterium tuberculosis complex by culture or nucleic acid testing or clinical diagnosis accompanied by treatment [10] and (2) having received tuberculosis treatment in Victoria. Patients who were notified before 1995 or after 2019, along with those with missing residential addresses, were excluded from the study.” This information is now presented in a new section, in lines 107-115.

Managing missing data 

We used a listwise deletion method when missing data contained residential addresses (participants were allocated neither to regional nor metropolitan areas) because our exposure of interest was regionality. When missing data did not contain the key variable (i.e., residential address), we utilised the pairwise deletion approach, which allowed us to retain data and reduce the possibility of selection bias.” This information is now presented in a new section, in lines 172-175.

Results

1. Your feedback: Line 160: description of exclusion of observations should be in the Methods section. Forty-five cases had no residential addresses recorded and were allocated neither to regional nor metropolitan areas and were excluded from the analysis. 

Our response: We have amended the manuscript to reflect this feedback. This information is now presented in a new section, in lines 195-197.

2. Your feedback: Line 164-167: again, description of exclusion of observations should be in the Methods section

Our response: Data recorded before 2005 had missing treatment outcomes for many cases and were excluded from the analysis of treatment outcomes but was included in other analyses. A statement on how missing data were managed has been added to the methods. This information is now presented in a new section, in lines 170-175.

3. Your feedback: Line 199-200: table 2, the numbers are not properly formatted. Please revise.

Our response: We have formatted the numbers.

4. Your feedback: Line 243: Seems multivariable and multivariate are used interchangeably. Please maintain uniformity, use multivariable.

Our response: We have amended the manuscript accordingly.

Discussion

1. Your feedback: Line 289: “The higher TB rate among the overseas-born people could be attributed to latent TB reactivation”

Comment: Authors need to add references to support this statement.

Our response: We have added the reference. The high TB notification within the first five years of arrival may be attributed to latent TB reactivation (15). Lines 348-349.

2. Your feedback: Authors should add a study limitation section

Our response: We have added the limitations in the discussion section of the manuscript.

“Limitations

Strengths of this study include the use of a comprehensive central database that includes important demographic, clinical and laboratory data, allowing for the incorporation of other factors outlined in this manuscript and a long study period of 25 years. However, we acknowledge that data for the entire study period are not available for all data fields (e.g., treatment outcomes, health system delays) due to data completeness, limiting trend analysis. Some of the data in our study, such as dates of symptom onset and healthcare presentation, were collected retrospectively from patients and thus may contain inaccuracies relating to recall bias. Data on some factors that could have influenced the treatment delay and outcomes, such as educational level was limited.” Lines 370-379.

---

## [Decision Letter · Decision Letter 1]

12 Aug 2022

PONE-D-22-00893R1Tuberculosis notifications in regional Victoria, Australia: implications for public health care in a low incidence settingPLOS ONE

Dear Dr. Moyo,

Thank you for submitting your manuscript to PLOS ONE. After careful consideration, we feel that it has merit but does not fully meet PLOS ONE’s publication criteria as it currently stands. Therefore, we invite you to submit a revised version of the manuscript that addresses the points raised during the review process.

We look forward to receiving your revised manuscript.

Kind regards,

Mohamed F. Jalloh, PhD, MPH

Academic Editor

PLOS ONE

Journal Requirements:

Reviewers' comments:

Reviewer's Responses to Questions

**Comments to the Author**

1. If the authors have adequately addressed your comments raised in a previous round of review and you feel that this manuscript is now acceptable for publication, you may indicate that here to bypass the “Comments to the Author” section, enter your conflict of interest statement in the “Confidential to Editor” section, and submit your "Accept" recommendation.

Reviewer #2: (No Response)

Reviewer #3: (No Response)

2. Is the manuscript technically sound, and do the data support the conclusions?

Reviewer #2: Yes

Reviewer #3: Partly

3. Has the statistical analysis been performed appropriately and rigorously? 

Reviewer #2: Yes

Reviewer #3: No

4. Have the authors made all data underlying the findings in their manuscript fully available?

Reviewer #2: Yes

Reviewer #3: Yes

5. Is the manuscript presented in an intelligible fashion and written in standard English?

Reviewer #2: Yes

Reviewer #3: Yes

6. Review Comments to the Author

Reviewer #2: Thank you so much for addressing the comments on the previous version of the manuscript. Below are my new comments based on the revised manuscript:

The Methods section still requires revisions. Currently there are sub-sections which are not properly described. For example, the study design sub-section is just one short sentence. I suggest revising this section to include the following sub-sections in this particular order: Data Source, Study Population, Variables, and Statistical Analysis. Under the Data Source sub-section - describe the data source, study design and study setting. Under Study Population sub-section - describe the target population by age, inclusion and exclusion criteria and missing data. Under the Variables sub-section - describe the variables, dependent and independent variables. Under Statistical Analysis sub-section - describe the analyses conducted in the study.

Reviewer #3: Lines 30-32: the methods section of the abstract is quite brief missing important details that will help readers to understand the results presented from a survival analysis.

Sample size, follow-up, date variables, outcomes, censoring, and statistical analysis issues are missing.

Line 36-39the statement in these lines feels like labeling. In the absence of adequate number of cases, it is difficult to associate multidrug resistant TB and being overseas born.

Furthermore, a statement in line 36 reads, 'the proportion of MDR-TB cases in regional vs metropolitan areas is similar'. In the next line, however, it presents only four cases of MDR-TB in regional vs 97 in metropolitan. The statements in the lines indicated above are difficult to follow�present them consistently in terms of proportion or in absolute numbers.

The data presented in Table 1 of the body of the document, do not support this statement. 4 MDR-TB vs 0 in regional and metropolitan this data is not adequately powered to support the statement provided in these lines.

Lines 40-44: "Cases with extra pulmonary TB in regional areas have a non-significantly longer healthcare system delay than patients in metropolitan (median 64 days versus 54 days, AHR = 0.8, 95% CI 0.6-1.0, P = 0.094). People living in regional areas have a non-significantly higher odds of dying of TB (AOR = 1.8, 95% CI 0.7-4.2, P = 0.198)."

In the above text, the authors presented mixed effect sizes, AHR vs AOR. However, the effect estimate from the Cox proportional hazards region is expressed in HRs than ORs. The other thing is that the authors should clearly specify their outcome of interest than generally providing 'TB mgt. outcome'.

The Background

Lines 56-74:

While information presented in the background is critical to understand context of the problem, its nature, efforts to reduce the extent of the problem, challenges, gaps, and the need to conduct the current study, it is only presented in 18 lines missing important details. Therefore, i suggest the authors to consider adding a few details to give insight to the problem studied.

Data analysis

Line 139: consider here too the comments provided in the abstract regarding data analysis

Lines 150-153: present the global test results and also for the independent variables to attest that the proportional hazard regression was met.

Results

Line 158: the 7% and 93% reported cases of TB do not reflect that 45 cases did not have residential information regarding their affiliation to regional of metropolitan.

Line 164-65: it is good to present the number of cases excluded. Or preferably provide the progress of pts. in a flow diagram.

Line 175-177: this result has not been well reflected in the abstract.

Table 1: the font size of contents of the table is significantly different from the text in the body. The authors also consider avoiding '-' in places where the total add to hundred or cell values added to the sample in respective subgroup. Or consider using '0' or NA to represent 'not available'

Discussion

Owning to the arrival of overseas born individuals from high TB burden countries to Australia, there could be an active search for TB among this particular group which may introduce a diagnostic suspicion bias. Was there an effort in this study to exclude that diagnostic suspicion bias was not an issue or was there an effort to reduce it if there was any?

7. PLOS authors have the option to publish the peer review history of their article (what does this mean?). If published, this will include your full peer review and any attached files.

Reviewer #2: No

Reviewer #3: No

---

## [Author Response · Author response to Decision Letter 1]

3 Sep 2022

Tuberculosis notifications in regional Victoria, Australia: implications for public health care in a low incidence setting

Response to reviewer 2

Your feedback 1. The Methods section still requires revisions. Currently there are sub-sections which are not properly described. For example, the study design sub-section is just one short sentence. I suggest revising this section to include the following sub-sections in this particular order: Data Source, Study Population, Variables, and Statistical Analysis. Under the Data Source sub-section - describe the data source, study design and study setting. Under Study Population sub-section - describe the target population by age, inclusion and exclusion criteria and missing data. Under the Variables sub-section - describe the variables, dependent and independent variables. Under Statistical Analysis sub-section - describe the analyses conducted in the study. 

Our response. We have organised the methods section under the suggested sub-sections. See revised lines 99 to 167.

Response to reviewer 3

Your feedback 1: Lines 30-32: the methods section of the abstract is quite brief missing important details that will help readers to understand the results presented from a survival analysis. Sample size, follow-up, date variables, outcomes, censoring, and statistical analysis issues are missing.

Our response. We have amended the abstract, and it now reads, “Background: Regionality is often a significant factor in tuberculosis (TB) management and outcomes worldwide. A wide range of context-specific factors may influence these differences and change over time. We compared TB treatment in regional and metropolitan areas, considering demographic and temporal trends affecting TB diagnosis and outcomes. Methods: Retrospective analyses of data for patients notified with TB in Victoria, Australia, were conducted. The outcomes were treatment delays and treatment outcomes. Multivariable Cox proportional hazard model analyses were performed to investigate the effect of regionality in the management of TB. Six hundred and eleven (7%) TB patients were notified in regional and 8,163 (93%) in metropolitan areas between 1995 and 2019. Of the 611 cases in the regional cohort, 401 (66%) were overseas-born. Fifty-one percent of the overseas-born patients in regional Victoria developed TB disease within five years of arrival in Australia. Four cases of multidrug-resistant tuberculosis were reported in regional areas, compared to 97 cases in metropolitan areas. A total of 3,238 patients notified from 2012 to 2019 were included in the survival analysis. Patient follow-up was censored at the first visit to the health care facility (Patient treatment delay) and at the initiation of TB treatment (Health system delay). Patient, health system, and treatment delays were similar in regional and metropolitan areas for cases with pulmonary involvement. Cases with extrapulmonary TB in regional areas have a non-significantly longer healthcare system delay than patients in metropolitan (median 64 days versus 54 days, AHR = 0.8, 95% CI 0.6-1.0, P = 0.094). 

Conclusion: Tuberculosis in regional Victoria is common among the overseas-born population, and patients with extrapulmonary TB in regional areas experienced a non-significant minor delay in treatment commencement with no apparent detriment to treatment outcomes. Improving access to LTBI management in regional areas may reduce the burden of TB.” Lines 26-49.

Your feedback 2. Line 36-39the statement in these lines feels like labeling. In the absence of adequate number of cases, it is difficult to associate multidrug resistant TB and being overseas born. Furthermore, a statement in line 36 reads, 'the proportion of MDR-TB cases in regional vs metropolitan areas is similar'. In the next line, however, it presents only four cases of MDR-TB in regional vs 97 in metropolitan. The statements in the lines indicated above are difficult to follow�present them consistently in terms of proportion or in absolute numbers. The data presented in Table 1 of the body of the document, do not support this statement. 4 MDR-TB vs 0 in regional and metropolitan this data is not adequately powered to support the statement provided in these lines.

Our response. We have amended the statement and it now reads “Four cases of multidrug-resistant tuberculosis were reported in regional areas, compared to 97 cases in metropolitan areas.” Lines 37-38.

Your feedback 3. Lines 40-44: "Cases with extra pulmonary TB in regional areas have a non-significantly longer healthcare system delay than patients in metropolitan (median 64 days versus 54 days, AHR = 0.8, 95% CI 0.6-1.0, P = 0.094). People living in regional areas have a non-significantly higher odds of dying of TB (AOR = 1.8, 95% CI 0.7-4.2, P = 0.198)."

In the above text, the authors presented mixed effect sizes, AHR vs AOR. However, the effect estimate from the Cox proportional hazards region is expressed in HRs than ORs. The other thing is that the authors should clearly specify their outcome of interest than generally providing 'TB mgt. outcome'.

Our response. We have amended the abstract, please refer to our response 1. Lines 26-49.

Your feedback 4. The other thing is that the authors should clearly specify their outcome of interest than generally providing 'TB mgt. outcome'.

Our response. We have added the following statement to the abstract and methods sections: “The study outcomes were treatment delays and treatment outcomes”. Lines 31, 144-145.

Your feedback 5. While information presented in the background is critical to understand context of the problem, its nature, efforts to reduce the extent of the problem, challenges, gaps, and the need to conduct the current study, it is only presented in 18 lines missing important details. Therefore, i suggest the authors to consider adding a few details to give insight to the problem studied.

Our response. Thank you for this suggestion. We have added the following statement: “Understanding TB treatment delays among regional patients provides important insights into Victorian TB programme performance and is a critical step towards tuberculosis elimination. Globally, TB surveillance data have been recognised as an important data source for assessing the disease burden and epidemiological trends in TB (World Health Organization, 2022). Evaluating treatment outcomes and delays in regional areas will inform practice and policy.” Lines 90-95.

Your feedback 6. Data analysis- Line 139: consider here too the comments provided in the abstract regarding data analysis. Lines 150-153: present the global test results and also for the independent variables to attest that the proportional hazard regression was met.

Our response. We have amended the data analysis, and it now reads: “Descriptive and multivariable analyses were performed. Incidence rates were calculated using the mid-year estimated resident population. Pearson’s x2 test was used to test the association between categorical variables. A two-tailed p-value of <0.05 was considered statistically significant. In logistic regression, we compared complete treatment with death, irrespective of the cause, lost to follow-up and transferred interstate or overseas. Died of TB was compared with completed treatment, lost to follow-up, died of other causes during treatment for TB, and transferred interstate or overseas. We included all independent variables in all multivariable analyses because we believed they could all affect the outcomes. However, our variable of interest was regionality. The proportional-hazards assumption was assessed using Kaplan-Meier survival curves by including time-dependent covariates in the model and with Schoenfeld residuals. In cases where proportionality assumptions were not met, analyses were stratified. Kaplan-Meier survival curves were used to show various delays in presentation, diagnosis, and treatment between regional and metropolitan cohorts, and Cox proportional hazard analyses were performed to assess these delays. Patient follow-up was censored, 1. at the first visit to a health care facility (Patient treatment delay), 2. at the initiation of TB treatment (Health system delay), 3. at the time a chest x-ray was performed (Diagnostic delay), 4. at the treatment initiation (Treatment initiation delay). Because of limited previous data, analyses of treatment outcomes were conducted using data from 2005 to 2019, while analyses of treatment delays used data from 2012 to 2019.” Lines 190-210.

Your feedback 7. Results- Line 158: the 7% and 93% reported cases of TB do not reflect that 45 cases did not have residential information regarding their affiliation to regional of metropolitan. 

Our response: We have amended this statement, and now it reads; “A total of 8,819 TB cases were notified to the Victorian Government Department of Health between 1995 and 2019. Among the 8,819 cases, 611 (7%) were recorded in regional areas, 8,163 (93%) in metropolitan areas of Victoria and 45 (1%) had neither regional nor metropolitan residential addresses (see Fig 1). Forty-five cases with no residential addresses were excluded from the study as they were classified as neither regional nor metropolitan.” Lines 215-219.

Your feedback 8. Line 164-65: it is good to present the number of cases excluded. Or preferably provide the progress of pts. in a flow diagram.

Our response. Thank you for this suggestion. We have now provided the flow of patients through the study, see Fig 1. Line 224.

Your feedback 9. Line 175-177: this result has not been well reflected in the abstract.

Our response. Lines 175-177 were referring specifically to issues related to missing data. However, the abstract has been amended to more clearly reflect the overall findings as outlined above. Lines 26-49.

Your feedback 10. Table 1: the font size of contents of the table is significantly different from the text in the body. 

Our response. We have increased the font size on all tables to 11 points.

Your feedback 11. The authors also consider avoiding in places where the total add to hundred or cell values added to the sample in respective subgroup. Or consider using '0' or NA to represent 'not available'

Our response. The tables have been amended accordingly. 

Your feedback 12. Discussion- Owning to the arrival of overseas born individuals from high TB burden countries to Australia, there could be an active search for TB among this particular group which may introduce a diagnostic suspicion bias. Was there an effort in this study to exclude that diagnostic suspicion bias was not an issue or was there an effort to reduce it if there was any?

Our response. Thanks for raising this point, which we agree may be important in many contexts. We do not believe that this is a significant issue in our study, as the considerable majority of migration-associated testing for TB an active case finding occurs prior to visa issuing in countries of origin and are thus not reflected in these TB cases presented here. Overall, then, we do not account further for diagnostic suspicion bias during the study period.

Reference

World Health Organisation. (2022). Strengthening TB surveillance. World Health Organisation. https://www.who.int/westernpacific/activities/strengthening-tb-surveillance

---

## [Decision Letter · Decision Letter 2]

3 Oct 2022

PONE-D-22-00893R2Tuberculosis notifications in regional Victoria, Australia: implications for public health care in a low incidence settingPLOS ONE

Dear Dr. Moyo,

Thank you for submitting your manuscript to PLOS ONE. After careful consideration, we feel that it has merit but does not fully meet PLOS ONE’s publication criteria as it currently stands. Therefore, we invite you to submit a revised version of the manuscript that addresses the points raised during the review process.

We look forward to receiving your revised manuscript.

Kind regards,

Mohamed F. Jalloh, PhD, MPH

Academic Editor

PLOS ONE

Journal Requirements:

Reviewers' comments:

Reviewer's Responses to Questions

**Comments to the Author**

1. If the authors have adequately addressed your comments raised in a previous round of review and you feel that this manuscript is now acceptable for publication, you may indicate that here to bypass the “Comments to the Author” section, enter your conflict of interest statement in the “Confidential to Editor” section, and submit your "Accept" recommendation.

Reviewer #2: All comments have been addressed

Reviewer #3: (No Response)

Reviewer #4: (No Response)

2. Is the manuscript technically sound, and do the data support the conclusions?

Reviewer #2: Yes

Reviewer #3: Yes

Reviewer #4: No

3. Has the statistical analysis been performed appropriately and rigorously? 

Reviewer #2: Yes

Reviewer #3: Yes

Reviewer #4: No

4. Have the authors made all data underlying the findings in their manuscript fully available?

Reviewer #2: Yes

Reviewer #3: Yes

Reviewer #4: Yes

5. Is the manuscript presented in an intelligible fashion and written in standard English?

Reviewer #2: Yes

Reviewer #3: Yes

Reviewer #4: No

6. Review Comments to the Author

Reviewer #2: (No Response)

Reviewer #3: Manuscript Number PONE-D-22-00893R2

I appreciate that the authors have considered the comments and improved the manuscript from the previous submission. However, I still request the authors to improve the background and discussion. In the background, the authors provided information that otherwise provided in the methods section under the subheading of study setting.

The authors have run two different models and identified four factors (three in the logistic model and one in the Cox model). However, they did not discuss these factors in detail (or not available) in the discussion section.

Pending the correction of the above comments, I recommend that the paper is accepted for publication.

Reviewer #4: Overall, the paper conducted a retrospective study analyzing Public Health Events Surveillance System data from 1995-2019 to understand the regionality differences on TB related treatment outcomes. The study added value to inform practice and policy.

Generally I have concerns on the integrity and transparency of the approaches, statistical analysis, and its write-up for the results.

• For any study, statistical analysis should serve the purpose of answering specific research questions. The authors did described the research questions (trend, and difference of regionality on outcomes), and statistical analysis conducted. But I do not see how the statistical approach were used to answer each question. For example, the author provided trend chart to describe the trend but failed to describe in the methods section/statistical analysis section that he did this. I did not realize the author used a trend chart until I read the results section. Secondly, the author in many places at the results section described “stable, increase, decrease” without statistical testing the data. Visually see the point estimation in a trend chart is not strong enough to make such conclusion. The author should at least add error bar in the fig and/or describe statistical tests results in the text.

o The authors described chi-square used but I did not see any chi-square results.

o It seems to me the author collapsed the age groups in to <65 and >65 in alter analysis, but this is not described anywhere. In fact, in table 1, I see a more granulated age category. This is not consistency throughout.

o Similar to the issued to the last bullet, the author collapsed multiple years of data when presenting the trend chart, it is fine to do so but the author did not describe this in any section.

o The flow chart should be included in the methods/study sample section rather than the results section

• The study did not clearly define what are the variables included in the study and why they are included. For example, died of TB clearly is not an independent variable, even if it is not the primary interested outcome (although I think they are important outcome variables). They should not be described as independent variables (line 164-166).

o The author described that they included all the covariates that are available in the multivariable models because “they could all affect the outcomes (line 198)”, which does not seem to be a sound approach, the authors should either find literature support or conducted bivariate analysis to confirm that those are covariates included

• For survival analysis specifically, I agree with one of the reviewers who reviewed this paper before “the methods section of the abstract is quite brief missing important details that will help readers to understand the results presented from a survival analysis. Sample size, follow-up, date variables, outcomes, censoring, and statistical analysis issues are missing”. For time to event analysis, especially survival analysis, at least the reader needs to know how the authors calculate the time interval, when is the starting point of the time interval and what censoring approaches used (i.e., right censoring, left censoring). How many have the event (outcome = 1, how many lost to follow-up). The author added some language to respond to this comment but I don’t thin it fully addressed the concern.

• One last concern I had is on table 1 where cells with small counts were presented. I am not know the specific rule in Australia for reporting patient data. But if you request/download any public health data from US CDC, you’ll notice that any cross-tabulated data would mask the cells with small counts. This is because by combining multiple PHIs, we can easily identify individual patients if it presents in the small enough cell. I would recommend the author either mask those cells or collapse them into less granulated categories.

7. PLOS authors have the option to publish the peer review history of their article (what does this mean?). If published, this will include your full peer review and any attached files.

Reviewer #2: No

Reviewer #3: **Yes: **MELKAMU MERID

Reviewer #4: **Yes: **Wenshu Li

---

## [Author Response · Author response to Decision Letter 2]

20 Dec 2022

PONE-D-22-00893R2

Tuberculosis notifications in regional Victoria, Australia: implications for public health care in a low incidence setting

Reviewer #3: 

Your feedback 1. I appreciate that the authors have considered the comments and improved the manuscript from the previous submission. However, I still request the authors to improve the background and discussion. In the background, the authors provided information that otherwise provided in the methods section under the subheading of study setting.

Our response. We have further amended the background and discussion sections of this manuscript to expand on context for the reader. The background section now reads: “Early diagnosis and treatment of tuberculosis are crucial to the effectiveness of TB control programmes (Yimer, et al., 2014). Previous studies have reported delayed TB diagnosis in countries with low incidence (Dale et al., 2017; Kelly et al., 2017; Labuda, 2022). For example, Labuda et al. (2022) conducted a study involving 21 patients and reported that eight (38%) had their TB diagnosis delayed by months. In a comparable study involving 34 patients with TB, Kelly et al. (2017) reported that 17 (50%) were diagnosed with other diseases, and the average time between admission and tuberculosis diagnosis was five days. 

Some studies have explored the factors associated with delays in TB treatment (Cai et al., 2015; Rattananupong et al., 2015; Williams et al., 2018). In a systematic review and meta-analysis of 45 studies, male patients and extended travel times/distances to the initial healthcare provider were associated with shorter patient and provider delays (Cai et al., 2015). In addition, unemployment, low income, haemoptysis, and positive sputum smears were associated with patient delay (Cai et al., 2015). Another study involving 133 patients reported that cough and hospital admission were associated with shorter health system delays, while age 65 or older was associated with longer delays (Williams et al., 2018).

In a retrospective study involving 239,532 patients, Wallace (2008) found that TB prevalence and trends were comparable in metropolitan and regional areas. In a similar study, including 16,784 patients, Abubakar reported that 45% of cases did not complete therapy in rural areas, compared to 26% of cases in urban areas.

TB incidence in Victoria remains low, with 436 TB cases notified in 2018, representing 6.9 cases per 100,000 population (Bright et al., 2020). In Australia, research to date has tended to focus on metropolitan areas, where case numbers typically predominate. Understanding TB treatment delays among regional patients provides important insights into VTP performance and is a critical step towards tuberculosis elimination. Globally, TB surveillance data have been recognised as an important data source for assessing the disease burden and epidemiological trends in TB (World health organisation, 2022). Evaluating treatment outcomes and delays in regional areas will inform practice and policy. We aimed to describe notified TB cases in regional Victoria from 1995 to 2019, including trends and outcomes over these 24 years.” Lines 56-75.

Your feedback 2. The authors have run two different models and identified four factors (three in the logistic model and one in the Cox model). However, they did not discuss these factors in detail (or not available) in the discussion section.

Our feedback. We have expanded on discussion regarding these points, including adding the following statement to the discussion section: “Our analysis indicates that regionality is not an independent determinant of dying of TB. This is inconsistent with a study by Mutembo et al. (29), which reported that rural locations in Zambia had a 70% higher risk of death; however these differences have occurred in the context of our higher resourced and lower incidence setting. We did identify that that older age and male sex were predictors of lower treatment completion in regional Victoria, which may allow for interventions in our context. In particular, awareness of lower rates of treatment completion in these groups can be considered in assessing the level of follow up support and medication supervision required for people with TB, and encourage targeted education and adherence support services to groups with observed lower rates of completion.” Lines 469-478.

 

Reviewer #4: Comments to the Author

Your feedback 1. The authors did described the research questions (trend, and difference of regionality on outcomes), and statistical analysis conducted. But I do not see how the statistical approach were used to answer each question. For example, the author provided trend chart to describe the trend but failed to describe in the methods section/statistical analysis section that he did this. I did not realize the author used a trend chart until I read the results section. 

Our response. The following statement has been added to the analysis section: “We compared the number of TB cases and the TB incidence rates in regional and metropolitan areas from 1 January 1995 to 31 December 2019. Trends over this period were displayed graphically. Data were considered in 5 year increments, with the total number of notifications per 5 year category serving as the denominator. Given the recognised connection between age and risk of tuberculosis, we also stratified case notification by age for examination of overall trends.” Lines 226-231.

Your feedback 2. Secondly, the author in many places at the results section described “stable, increase, decrease” without statistical testing the data. Visually see the point estimation in a trend chart is not strong enough to make such conclusion. The author should at least add error bar in the fig and/or describe statistical tests results in the text.

Our response. Figure 2 reports the number of reported cases, there is no inferring. We are not intending to show significance, just commenting on the general trends observed in the graph. We have changed the statement and, it now reads “In the overseas-born cohort, 29 people were under the age of 20, 236 were between the ages of 20 and 49, and 136 were �50 (Fig 2). Twenty cases were under the age of 20, 43 were aged 20–49 years, and 137 were �50 in the Australian-born cohort. TB cases among overseas-born people aged 20-49 years increased from 22 to 79 cases between 2000 and 2019. Conversely, Australian-born cases in this age bracket remained effectively unchanged over the same period (11 cases vs 10 cases).” Lines 321-326.

Your feedback 3. The authors described chi-square used but I did not see any chi-square results.

Our feedback. The statement “Pearson’s x2 test was used to test the association between categorical variables” was deleted. Line 234.

 Your feedback 4. It seems to me the author collapsed the age groups in to <65 and >65 in alter analysis, but this is not described anywhere. In fact, in table 1, I see a more granulated age category. This is not consistency throughout.

Our response. We added the following statement to the analysis section: “We chose a more granular age category in Table 1 to describe the patients to determine which age group is impacted more by tuberculosis in order to plan for the targeted TB management program. The age groups < 65 years and � 65 years were used in the univariate and multivariate analysis because, in some studies, the age groups � 65 years have been found to be associated with poor treatment outcomes (Vasankari et al., 2007; Kang et al., 2021).” Lines 229-233.

Your feedback 5. Similar to the issued to the last bullet, the author collapsed multiple years of data when presenting the trend chart, it is fine to do so but the author did not describe this in any section.

Our feedback. We have addressed this point in response to feedback 1. Lines 224-228.

Your feedback 6. The flow chart should be included in the methods/study sample section rather than the results section

Our feedback. We followed the Strengthening The Reporting of Observational Studies in Epidemiology (STROBE) guidelines. According to the guidelines the flow chart is in results section (Von Elm, et al., 2007). Line 275.

Your feedback 7. The study did not clearly define what are the variables included in the study and why they are included. For example, died of TB clearly is not an independent variable, even if it is not the primary interested outcome (although I think they are important outcome variables). They should not be described as independent variables (line 164-166).

Our response. We amended the variable section, and the new section reads: 

“Variables

The variables in this study included patient outcomes and predictors of the outcomes. The study outcomes were dependable variables: treatment outcomes and treatment delays. Treatment outcomes included completed treatment, lost to follow-up, died of TB, died of other causes during treatment for TB, and being transferred interstate or overseas. 

Patients were defined as having died of TB when the clinical mode of death, the severity of TB disease (based on the presence of systemic symptoms, the extent of involvement of affected organs and central nervous system involvement), the presence of massive haemoptysis or respiratory, multisystem, or specific vital organ failure could be linked to tuberculosis and no other likely cause (Walpola, 2003). Died from other causes was defined as a death that was not attributed to tuberculosis based on pathological or autopsy examinations and death certificates indicating that the cause of death was another disease (Walpola, 2003). Patients were defined as having completed treatment if they had completed a minimum of 6 months of therapy and assessed as having completed treatment by the treating physician (Dale et al., 2016). Lost to follow-up was a treatment outcome for patients who did not complete therapy because they could not be located (Jasmer et al., 2004). Because of a change of residency, some patients were transferred interstate or overseas to continue their treatment. 

For the treatment delays, we adapted the definitions outlined by Van Wyk et al. (2011), which proposed that: ‘Patient treatment delay’ is the period (in the number of days) between the onset of any self-reported TB symptoms and the first visit to a health care facility. ‘Health system delay’ is the period (in the number of days) between the first healthcare facility visit and initiation of TB treatment. ‘Diagnostic delay’ is defined as the period (in the number of days) between the onset of any self-reported TB-related symptoms and the time a chest x-ray was performed. ‘Treatment initiation delay’ was the period between a positive specimen (TB confirmed) and treatment initiation. 

We extracted the following independent variables: (1) demographic data: age (age groups in years), sex (male or female), country of birth (name of the country), Aboriginal and Torres Strait Islander status (Aboriginal and/or Torres Strait Islander or not Aboriginal and/or Torres Strait Islander), local government areas (local government area), self-reported residency status (Australian-born, permanent resident, refugee/humanitarian, visitor, overseas student, other and unknown status), and for overseas-born cases, year of arrival in Australia (year). (2) Clinical characteristics: year of tuberculosis notification (year), the manifestation of tuberculosis (pulmonary, extrapulmonary or both), chest X-ray results (abnormal, cavitation or normal), laboratory results (smear, culture, or gene expert). Most of these variables have been reported as influencing patient outcomes in some studies (Dale et al., 2016; Kang et al., 2021). Lines 166-199.

 Your feedback 8. The author described that they included all the covariates that are available in the multivariable models because “they could all affect the outcomes (line 198)”, which does not seem to be a sound approach, the authors should either find literature support or conducted bivariate analysis to confirm that those are covariates included

Our feedback. We added the following statement to the analysis section: Area of residence, age, sex, drug susceptibility, and country of birth were included in univariable as well as multivariable analysis because there were found to affect TB treatment outcomes in some studies (Dale et al., 2016; Kang et al., 2021; Mukherjee et al., 2012). Lines 239-241.

Your feedback 9. For survival analysis specifically, I agree with one of the reviewers who reviewed this paper before “the methods section of the abstract is quite brief missing important details that will help readers to understand the results presented from a survival analysis. Sample size, follow-up, date variables, outcomes, censoring, and statistical analysis issues are missing”. For time to event analysis, especially survival analysis, at least the reader needs to know how the authors calculate the time interval, when is the starting point of the time interval and what censoring approaches used (i.e., right censoring, left censoring). How many have the event (outcome = 1, how many lost to follow-up). The author added some language to respond to this comment but I don’t thin it fully addressed the concern. 

Our response: We added the following statement to the abstract: “The time follow-up for patient delay started at symptom onset date, and the event was the presentation to the healthcare centre. For healthcare system delay, follow-up time started at the presentation to the healthcare centre, and the event was, commenced on TB treatment.” Lines 39-42.

We added the following statement to the analysis section: “We described the follow-up periods as follows: The time follow-up for patient delay started at symptom onset date, and the event was the presentation to the healthcare centre. For healthcare system delay, follow-up time started at the presentation to the healthcare centre, and the event was, commenced on TB treatment. Diagnostic delay: follow-up time began at the presentation to the healthcare centre, and the event was the first chest x-ray examination. Treatment delay: follow-up time started at first abnormal chest X-ray examination, and the event was, commenced on TB treatment. Treatment delay 2: follow-up began on the first date the patient had a positive specimen for TB, and the event, was commenced on TB treatment. Censoring occurred when patients failed to have an event because they either transferred interstate/overseas or dies or lost to follow-up (right censor).” Line 253-263.

Your feedback 10: How many have the event (outcome = 1, how many lost to follow-up). The author added some language to respond to this comment but I don’t thin it fully addressed the concern. 

Our feedback. We have added the following information and a table to the results section: “

Table 5 shows patients who did not have an event in the survival analysis. Among 1,343 patients with pulmonary TB in the metropolitan area, 161 did not undergo chest x-ray examination after their first presentation to the health care centre compared to 16 out of 122 in the rural area. Nine out of 1,107 people in metropolitan with extrapulmonary TB did not commence treatment after their first presentation to the healthcare centre compared to none in rural areas. The main reasons for not having an event in the treatment delay analysis were lost to follow-up, died of another cause, or died of TB, or transferred interstate or overseas before the event. For example, in health system delay for patients with pulmonary involvement, in the rural cohort, one patient died from another cause, two died of TB and one transferred interstate or overseas after first presenting to the health care centre but before commencing TB therapy. In the metropolitan cohort, one was lost to follow-up, eight died from another cause, 12 died of TB, and 17 transferred interstate or overseas before commencing treatment.”

Table 5. Patients who did not have an event in the survival analysis

 Patients who did have an event Patients who did not have an event Patients who did have an event Patients who did not have an event

Patients with pulmonary involvement Rural Metropolitan 

Patient delay. Event: attending health centre after experiencing TB symptoms 113 0 1,293 0

Health system delay. Event: starting treatment after first health care centre visit 146 4 1,691 38

Diagnosis delay. Event: having chest x-ray after the first presentation to the health care centre 122 16 1,343 161

Treatment delay: Event: starting tuberculosis treatment after undergoing first chest x-ray 131 3 1,565 21

Extrapulmonary patients 

Patient delay. Event: attending health centre after experiencing TB symptoms 62 0 907 0

Health system delay. Event: starting treatment after first health care centre visit 78 0 1,107 9

Diagnosis delay. Event: having chest x-ray after the first presentation to the health care centre 47 22 798 218

Treatment delay 2. Event: treatment initiation after first positive TB specimen test 151 7 1,996 51

Your feedback 11. One last concern I had is on table 1 where cells with small counts were presented. I am not know the specific rule in Australia for reporting patient data. But if you request/download any public health data from US CDC, you’ll notice that any cross-tabulated data would mask the cells with small counts. This is because by combining multiple PHIs, we can easily identify individual patients if it presents in the small enough cell. I would recommend the author either mask those cells or collapse them into less granulated categories.

Our feedback: Given that the data presented in Table 1 covers a wide geographic region and a 25 year period of time, we consider that the risk of patient identification in this data set is low. To avoid any perceived issues with confidentiality, after review of this table we have collapsed the row for contemporaneous treatment (n=1) into the ‘unknown outcome’ category.

 

Reference:

Abubakar, I., Crofts, J. p., Gelb, D., Story, A., Andrews, N., & Watson, J. m. (2007). Investigating urban–rural disparities in tuberculosis treatment outcome in England and Wales. Epidemiology and Infection, 136(1), 122–127. https://doi.org/10.1017/S0950268807008333

Bright, A., Denholm, J., Coulter, C., Waring, J., & Stapledon, R. Tuberculosis notifications in Australia, 2015-2018. Commun Dis Intell. 2020;44:1–39. 

Cai, J., Wang, X., Ma, A., Wang, Q., Han, X., Li, Y. (2015). Factors associated with patient and provider delays for tuberculosis diagnosis and treatment in asia: a systematic review and meta-analysis. PLoS ONE [Electronic Resource], 10, e0120088. https://dx.doi.org/10.1371/journal.pone.0120088

Dale, K., Tay, E., Trauer, J. M., Trevan, P., & Denholm, J. (2017). Gender differences in tuberculosis diagnosis, treatment and outcomes in Victoria, Australia, 2002-2015. The International Journal of Tuberculosis and Lung Disease, 21(12), 1264–1271. https://doi.org/10.5588/ijtld.17.0338

Dale, K., Tay, E., Trevan, P., & Denholm, J. T. (2016). Mortality among tuberculosis cases in Victoria, 2002-2013: case fatality and factors associated with death. The International Journal of Tuberculosis and Lung Disease, 20(4), 515–523. https://doi.org/10.5588/ijtld.15.0659

Jasmer, R. M., Seaman, C. B., Gonzalez, L. C., Kawamura, L. M., Osmond, D. H., & Daley, C. L. (2004). Tuberculosis Treatment Outcomes: Directly Observed Therapy Compared with Self-Administered Therapy. American Journal of Respiratory and Critical Care Medicine, 170(5), 561–566. https://doi.org/10.1164/rccm.200401-095OC

Kang, Y., Jo, E. J., Eom, J. S., Kim, M. H., Lee, K., Kim, K. U., Park, H. K., Lee, M. K., & Mok, J. (2021). Treatment Outcomes of Patients with Multidrug-Resistant Tuberculosis: Comparison of Pre- and Post-Public-Private Mix Periods. Tuberculosis and respiratory diseases, 84(1), 74–83. https://doi.org/10.4046/trd.2020.0093

Kelly, A. M., D'Agostino, J. F., Andrada, L. V., Liu, J., Larson, E. (2017). Delayed tuberculosis diagnosis and costs of contact investigations for hospital exposure: new york city, 2010-2014. American Journal of Infection Control, 45, 483-486. https://dx.doi.org/10.1016/j.ajic.2016.12.017

Mukherjee, A., Saha, I., Sarkar, A., & Chowdhury, R. (2012). Gender differences in notification rates, clinical forms and treatment outcome of tuberculosis patients under the RNTCP. Lung India, 29(2), 120-122. doi:https://doi.org/10.4103/0970-2113.95302

Mutembo, S., Mutanga, J. N., Musokotwane, K., Kanene, C., Dobbin, K., Yao, X., Li, C., Marconi, V. C., & Whalen, C. C. (2019). Urban-rural disparities in treatment outcomes among recurrent TB cases in Southern Province, Zambia. BMC Infectious Diseases, 19(1), 1087–1087. https://doi.org/10.1186/s12879-019-4709-5

Rattananupong, T., Hiransuthikul, N., Lohsoonthorn, V., Chuchottaworn, C. (2015). Factors associated with delay in tuberculosis treatment at 10 tertiary level care hospitals in thailand. Southeast Asian Journal of Tropical Medicine & Public Health, 46(4), 689-96. Retrieved from http://ovidsp.ovid.com/ovidweb.cgi?T=JS&PAGE=reference&D=med12&NEWS=N&AN=26867389.

Van Wyk SS, Enarson DA, Beyers N, Lombard C, Hesseling AC. Consulting private health care providers aggravates treatment delay in urban South African tuberculosis patients. Int J Tuberc Lung Dis. 2011;15(8):1069–76. Imperial, M. Z., Nahid, P., Phillips, P. P., Davies, G. R., Fielding, K., Hanna, D., ... & Savic, R. M. (2018). A patient-level pooled analysis of treatment-shortening regimens for drug-susceptible pulmonary tuberculosis. Nature medicine, 24(11), 1708-1715.

 Vasankari, T., Holmström, P., Ollgren, J., Liippo, K., Kokki, M., & Ruutu, P. (2007). Risk factors for poor tuberculosis treatment outcome in Finland: a cohort study. BMC Public Health, 7(1), 291–291. https://doi.org/10.1186/1471-2458-7-291

Von Elm, E., Altman, D. G., Egger, M., Pocock, S. J., Gøtzsche, P. C., & Vandenbroucke, J. P. (2007). The Strengthening the Reporting of Observational Studies in Epidemiology (STROBE) Statement: Guidelines for Reporting Observational Studies. Epidemiology (Cambridge, Mass.), 18(6), 800–804. https://doi.org/10.1097/EDE.0b013e3181577654

Wallace, R. M., Armstrong, L. R., Pratt, R. H., Kammerer, J. S., & Iademarco, M. F. (2008). Trends in Tuberculosis Reported From the Appalachian Region: United States, 1993-2005. The Journal of Rural Health, 24(3), 236–243. https://doi.org/10.1111/j.1748-0361.2008.00164.x

 Walpola, H. C., Siskind, V., Patel, A. M., Konstantinos, A., & Derhy, P. (2003). Tuberculosis-related deaths in Queensland, Australia, 1989-1998: characteristics and risk factors. The International Journal of Tuberculosis and Lung Disease, 7(8), 742–750.

 Williams, E., Cheng, A. C., Lane, G. P., & Guy, S. D. (2018). Delays in presentation and diagnosis of pulmonary tuberculosis: a retrospective study of a tertiary health service in Western Melbourne, 2011–2014. Internal Medicine Journal, 48(2), 184–193. https://doi.org/10.1111/imj.13551

World Health Organisation. Strengthening TB surveillance. [Internet]. 2022 [cited 2022 Jun 26]. Available from: https://www.who.int/westernpacific/activities/strengthening-tb-surveillance

Yimer, S. A., Bjune, G. A., & Holm-Hansen, C. (2014). Time to first consultation, diagnosis and treatment of TB among patients attending a referral hospital in Northwest, Ethiopia. BMC infectious diseases, 14, 19. https://doi.org/10.1186/1471-2334-14-19

---

## [Decision Letter · Decision Letter 3]

27 Feb 2023

Tuberculosis notifications in regional Victoria, Australia: implications for public health care in a low incidence setting

PONE-D-22-00893R3

Dear Dr. Moyo,

We’re pleased to inform you that your manuscript has been judged scientifically suitable for publication and will be formally accepted for publication once it meets all outstanding technical requirements.

Kind regards,

Mohamed F. Jalloh, PhD, MPH

Academic Editor

PLOS ONE

Additional Editor Comments (optional):

Reviewers' comments:

Reviewer's Responses to Questions

**Comments to the Author**

1. If the authors have adequately addressed your comments raised in a previous round of review and you feel that this manuscript is now acceptable for publication, you may indicate that here to bypass the “Comments to the Author” section, enter your conflict of interest statement in the “Confidential to Editor” section, and submit your "Accept" recommendation.

Reviewer #4: All comments have been addressed

2. Is the manuscript technically sound, and do the data support the conclusions?

Reviewer #4: Yes

3. Has the statistical analysis been performed appropriately and rigorously? 

Reviewer #4: Yes

4. Have the authors made all data underlying the findings in their manuscript fully available?

Reviewer #4: Yes

5. Is the manuscript presented in an intelligible fashion and written in standard English?

Reviewer #4: No

6. Review Comments to the Author

Reviewer #4: The author responded to reviewers comments and made methods section and statistical approach description more accurate and clear. But I do think there are some room for improvements. For example, the author responded to reviewers comments on rational to include covariates -"Most of these variables have been reported as influencing patient outcomes in some studies", the better way to do this is clearly describing what variables are associated with which outcome in previous study and state rational why some other variables are included. Overall, I think the author well addressed reviewer's comments.

7. PLOS authors have the option to publish the peer review history of their article (what does this mean?). If published, this will include your full peer review and any attached files.

Reviewer #4: No

---

## [Editor Report · Acceptance letter]

13 Mar 2023

PONE-D-22-00893R3 

Tuberculosis notifications in regional Victoria, Australia: implications for public health care in a low incidence setting 

Dear Dr. Moyo:

I'm pleased to inform you that your manuscript has been deemed suitable for publication in PLOS ONE. Congratulations! Your manuscript is now with our production department. 

Kind regards, 

on behalf of

Dr. Mohamed F. Jalloh 

Academic Editor

PLOS ONE